# Measuring the burden of hundreds of BioBricks defines an evolutionary limit on constructability in synthetic biology

Noor Radde[1,3], Genevieve A. Mortensen [1,3], Diya Bhat [1], Shireen Shah[1], Joseph J. Clements[1], Sean P. Leonard [1], Matthew J. McGuffie[1], Dennis M. Mishler[1,2] & Jeffrey E. Barrick [1] ✉

Engineered DNA will slow the growth of a host cell if it redirects limiting resources or otherwise interferes with homeostasis. Escape mutants that alleviate this burden can rapidly evolve and take over cell populations, making genetic engineering less reliable and predictable. Synthetic biologists often use genetic parts encoded on plasmids, but their burden is rarely characterized. We measured how 301 BioBrick plasmids affected *Escherichia coli* growth and found that 59 (19.6%) were burdensome, primarily because they depleted the limited gene expression resources of host cells. Overall, no BioBricks reduced the growth rate of *E. coli* by >45%, which agreed with a population genetic model that predicts such plasmids should be unclonable. We made this model available online for education (https://barricklab.org/burden-model) and added our burden measurements to the iGEM Registry. Our results establish a fundamental limit on what DNA constructs and genetic modifications can be successfully engineered into cells.

Synthetic biologists are engineering increasingly sophisticated functions into cells and deploying these living machines in new and more challenging environments. For example, cells have been created with genetic circuits that perform complex sensing and logic operations[1,2], and bacterial symbionts have been engineered to improve the productivity and health of their plant and animal hosts[3–5]. However, unlike computer code, engineered DNA sequences in cells can evolve, potentially making their functions unpredictable and unreliable[6,7]. Evolutionary failure—when less-functional or nonfunctional mutants outcompete their ancestor—can occur rapidly if an engineered function is highly burdensome to a cell or if the sequences that encode it are especially mutation-prone[8–12]. In extreme cases, a population of cells may already become dominated by escape mutants that have evolved inactivated variants of a designed sequence after the outgrowth of a single transformed cell into a colony or small laboratory culture, making that construct essentially unclonable. To improve the foundations of bioengineering, we need to better understand why certain DNA constructs are more burdensome to cells than others and the limits on how much burden a cell can tolerate before unwanted evolution becomes a barrier.

Because engineered DNA constructs use resources from the cell to replicate and express genes, these processes are the most common and predictable sources of burden[13]. Transcriptional resources (e.g., RNA polymerases) or translational resources (e.g., ribosomes, charged tRNAs) often become limiting when a foreign DNA construct directs the synthesis of RNAs and proteins that are not native to the cell. Protein overexpression studies in *E. coli* generally find that ribosomes are the most limiting factor. Translating these foreign proteins decreases the growth rates of cells in proportion to how many ribosomes are redirected away from producing host proteins[14–18]. Usage of gene expression resources can be monitored using high-throughput approaches that globally profile RNA abundance and ribosomal occupancy[19,20] or reporter genes with expression levels that reflect the

[1]Department of Molecular Biosciences, Center for Systems and Synthetic Biology, The University of Texas at Austin, Austin, TX, USA. [2]The Freshman Research Initiative, College of Natural Sciences, The University of Texas at Austin, Austin, TX, USA. [3]These authors contributed equally: Noor Radde, Genevieve A. Mortensen. ✉e-mail: jbarrick@cm.utexas.edu

depletion of overall cellular capacities for transcription and translation[21].

Burden may also arise due to how specific gene products expressed from an engineered DNA construct interact with host cells. Metabolic engineering purposefully funnels precursor molecules toward a target compound by expressing foreign enzymes, altering gene regulation, and/or disrupting native pathways. These modifications will generally slow a cell's growth, and metabolic products or intermediates may also accumulate to levels that are detrimental to cellular physiology[22–24]. Expressing certain types of proteins, such as proteases and integral membrane proteins, is also known to be stressful or toxic to *E. coli* cells, due either directly to their functions or to competition with native proteins for secretion machinery[25,26]. Proteins used for orthogonal control of gene expression, like T7 RNA polymerase and dCas9, can exhibit excessive activity or off-target effects that are extremely burdensome[27]. Finally, unintentional expression of antisense and frameshifted gene products from cryptic promoters and ribosome-binding sites has been shown to be an unexpected source of burden in some constructs[9,20].

Sharing of standardized genetic parts has been a cornerstone of synthetic biology since its inception[28,29]. The Registry of Standard Biological Parts is a database of engineered DNA sequences[30] that thousands of teams have contributed to as part of their participation in the International Genetically Engineered Machines (iGEM) competition[31,32]. Most BioBrick parts are cloned into a small set of standard vector backbones, which makes these plasmids a useful common garden for analyzing the properties of inserts encoding different genetic parts and devices. In past studies, BioBricks have been used to compare standardized measurements of promoter strength[33] and fluorescent protein expression[34,35] across many labs. It has been proposed that genetic reliability—in the evolutionary sense of how many cell doublings it takes for an engineered function to decay within a population—be listed on a data sheet describing a genetic part[29], but this property is rarely characterized in practice. One goal of iGEM is to improve upon existing parts, and many BioBrick sequences are reused by synthetic biology researchers outside of iGEM. Therefore, characterizing which of these parts are evolutionarily unstable and understanding why this is the case would broadly benefit the field.

We measured the burden of 301 BioBrick plasmids from the iGEM Registry containing DNA constructs ranging from individual parts to complex devices. None of these plasmids reduced the growth rate of their *E. coli* hosts by >45%, in agreement with stochastic simulations of evolution that predict a level of burden above this threshold would make a construct unclonable. We found that 6 BioBrick plasmids had a burden of >30%, which would be expected to be problematic on the laboratory scale, and that 19 had a burden of >20%, enough that they might fail during process scale-up or in other applications in which cells continue to divide. Several BioBrick plasmids, including two we used as controls, evolved mutations that likely reduce their burden by compromising their designed functions. Finally, we determined that depletion of gene expression resources is sufficient to explain the burden of most BioBrick plasmids, though some reduce host growth rates for other, currently unknown reasons. Our work demonstrates standardized frameworks for measuring burden and simulating the dynamics of evolutionary failure that can be used to improve the reliability of bioengineering.

## Results

### Model of evolutionary failure

Growth of a cell population that has been engineered with a new DNA construct begins from a single transformed cell. As the population divides, progeny with mutations in the sequence of the designed DNA construct will arise. If these mutations alleviate a burden on the cells caused by the engineered DNA—most often by lessening or eliminating a designed function that compromises their growth—then, the mutant

cells will have a competitive advantage. These higher-fitness cells will outreplicate and displace ancestral cells with the original DNA construct until they dominate within the population and function declines.

To put our experimental measurements of burden into context, we first investigated the expected timing of evolutionary failure using a differential equation model (Fig. 1A). This model has two parameters. The first is the burden ($b$) of the engineered DNA, expressed as a percent reduction in the rate of replication of a cell containing the genetic construct. The model makes a simplifying assumption that

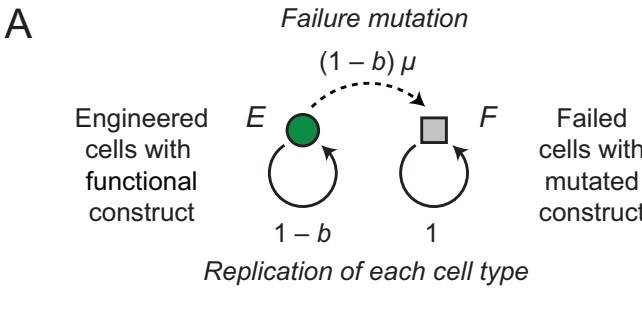

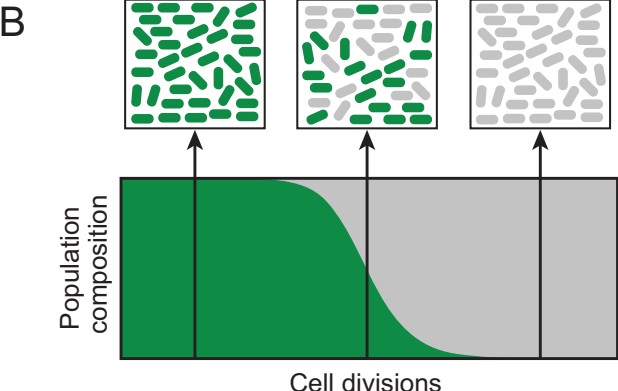

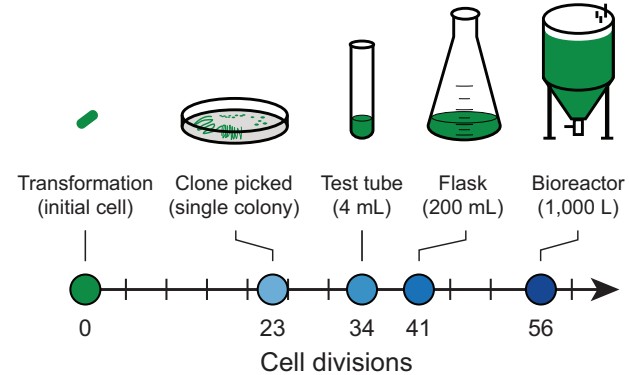

**Fig. 1 | Evolutionary failure of a population of engineered cells. A** Graphical representation of a differential equation model with one class of failure mutations that completely alleviates the fitness burden of an engineered DNA construct on a host cell. **B** Population dynamics expected from this model. Subpopulations of failed cells with mutated constructs evolve and outcompete the original engineered cells with functional constructs. Complete failure happens rapidly once the mutant cells reach a detectable frequency in the population. **C** Approximate numbers of cell divisions that occur as a single engineered cell is grown into cultures on different laboratory and industrial scales after its creation (see Supplementary Data 1). Source data are provided with this paper.

there is one category of mutations that leads to the failure of the engineered function in a way that completely alleviates its burden. The rate of these failure mutations per cell division ($\mu$) is the second parameter. The typical dynamics for this model are that broken cells with a failure mutation are initially very rare but then rapidly take over a population as their fitness advantage is exponentially compounded over time (Fig. 1B).

We wanted to understand what magnitude of burden would be likely to lead to evolutionary failure of an engineered function during a typical scale-up process starting with a single bacterial cell picked as a colony isolate after transformation with a newly cloned plasmid or some other genome editing procedure (Fig. 1C, Supplementary Data 1). We estimate that ~23 cell divisions occur by the time a single cell produces a normal-sized colony containing ~8 million cells on an agar plate. If this entire colony is placed into ~4 ml of LB in a test tube, it takes an additional ~11 cell divisions to reach saturation, assuming a final density of ~$5 \times 10^9$ cells/ml. Growth to a 200 mL laboratory scale at a higher cell density (e.g., in terrific broth for recombinant protein overexpression) brings the total to ~40 cell divisions. Larger-scale industrial processes can reach even higher cell densities such that ~56 cell divisions may be needed to saturate a 1000 L bioreactor.

The rates of mutations leading to the failure of different DNA constructs can vary widely, so we tested values of this parameter spanning several orders of magnitude: from $10^{-4}$ to $10^{-8}$ per genome per cell division. One factor that affects mutation rates is the information content of a sequence, i.e., how many base pairs must be specified to encode its function. Longer engineered DNA sequences and those that are less robust to base changes are at a greater risk for inactivating mutations[6,7]. The rate of base substitutions in *E. coli* is ~$5 \times 10^{-10}$ per base pair per generation[36,37], and most microbes with DNA genomes have similar mutation rates[38,39]. Thus, if a sequence contains protein-coding genes that constitute 1000 base pairs and 20% of the substitutions in these genes lead to a loss of function, the failure rate will be ~$1 \times 10^{-7}$ per cell division just from base substitutions. This estimate does not account for the possibility that there are mutational hotspots in a sequence, such as mononucleotide repeats, that can cause certain mutations to occur at much higher rates[40,41]. Furthermore, selfish elements in the host genome usually contribute other types of mutations that further increase the total rate of failure mutations. In particular, transposon insertions often inactivate genes or sequences required for gene expression in engineered DNA constructs[11,42,43].

In the end, empirical measurements generally find a rate of ~$10^{-6}$ per cell division for mutations that inactivate a single gene that is located in the chromosome of *E. coli* or another bacterium[42,44]. The effective mutation rate is much higher for engineered constructs maintained on multicopy plasmids because each copy of the plasmid in a cell is at risk. If there are 100 copies of a plasmid, the chance of a plasmid with a certain mutation arising is ~100-fold higher. So, for example, the rate of mutations reverting a stop codon in a reporter construct, which can only occur via one or a few single base substitutions, has been measured as ~$10^{-7}$ per cell division rather than the value of ~$10^{-9}$ expected if this reporter were tested in the chromosome[12]. For plasmids that lack partitioning systems like pBR322 and pUC derivatives commonly used in *E. coli*, one broken plasmid copy can rapidly lead to 100% failure of all plasmids in all cells in a population because progeny that happen to inherit more broken plasmid copies due to random segregation will outcompete those that do not. In summary, the effective rate of failure mutations in a high-copy plasmid is usually much higher than the point mutation rate; it is expected to be at least on the order of $10^{-5}$ and often as high as $10^{-4}$ per cell division. Though mutational hotspots and multicopy plasmid replication are not explicitly accounted for in our model, they justify exploring simulations with a wide range of mutation rates.

Previous studies of escape mutations have used the deterministic results of ordinary differential equation (ODE) models to estimate the times to failure of engineered cells[6,11]. This framework assumes that mutants appear continuously and immediately at the beginning of the simulation. However, in reality, mutations appear stochastically in single cells at very low rates, and the dynamics can vary greatly depending on whether these events occur early or late in the growth of a population. Therefore, we compared the deterministic results for our ODE model to stochastic simulations of this model to evaluate how and when the results varied. We found that deterministic simulations consistently overestimate how unstable a construct will be for a given combination of parameters compared to stochastic models (Fig. 2). The discrepancy becomes larger at lower mutation rates where it mainly reflects the waiting time needed for the rare event that generates the first mutant cell to appear in a population in the stochastic simulations, compared to the immediate appearance of these mutants in the deterministic simulations. However, there are also occasional stochastic simulation runs in which failure occurs sooner than it does in the deterministic model due to jackpot events when a mutation occurs early in the growth of a population (as seen in the panel for $b = 20\%$, $\mu = 10^{-8}$).

Because we expect it to better represent the true evolutionary dynamics, we further examined the results of the stochastic simulations (Fig. 3). They show that at a typical mutation rate of $10^{-5}$ per cell doubling (expected for a plasmid-borne construct) a burden of ≥50% would lead to takeover of broken mutants in a test-tube culture most of the time. At a mutation rate of $10^{-4}$, constructs with a burden of ≥40% would not survive on this small scale. Since one needs to grow a single transformed cell into a culture of this size to purify and sequence a plasmid to verify that it has the designed sequence, the model predicts that constructs this burdensome will be essentially unclonable. Even for less-burdensome plasmids or for constructs experiencing lower mutation rates (for example, single-copy genes in the chromosome), the model predicts that failure may occur at larger scales if the burden reaches the 20−30% range.

We created an online version of our model that allows users to adjust the burden and failure mutation rate parameters (https://barricklab.org/burden-model). There is an option to use the stochastic or deterministic version of the model and compare the results. Additionally, users can change the effective volume and density of their culture to understand the scale at which a DNA construct with certain characteristics is likely to fail. This interactivity allows users to explore a range of parameters and rerun simulations multiple times to see for themselves how mutations can affect the functional lifetimes of devices constructed in living, and therefore evolving cells.

## Burden of BioBrick parts

To test whether actual engineered DNA sequences obey the evolutionary constraints predicted by our model of escape mutations, we examined a diverse collection of engineered DNA sequences created for the iGEM (International Genetically Engineered Machine) competition[31]. These BioBricks range in complexity from small DNA parts, such as promoters and protein tags, to larger devices that consist of multiple genes and operons. Historically, BioBricks in the Registry of Standard Biological Parts had to be cloned into plasmids in ways that allowed them to be combined into larger constructs using a specific assembly standard[45]. As a consequence, most BioBricks in the kit distributed to iGEM teams are provided in plasmids pSB1C3, pSB1A2, or in both of these backbones (Fig. 4A). pSB1C3 and pSB1A2-share the same high-copy pUC origin of replication and overall organization, but they are maintained using different antibiotic resistance genes: chloramphenicol acetyltransferase (*cat*) which confers chloramphenicol resistance ($\text{Cam}^R$) for pSB1C3 versus β-lactamase (*bla*) which confers resistance to ampicillin and other β-lactams ($\text{Amp}^R$) for pSB1A2. These plasmids also differ in how the expression of the cloned

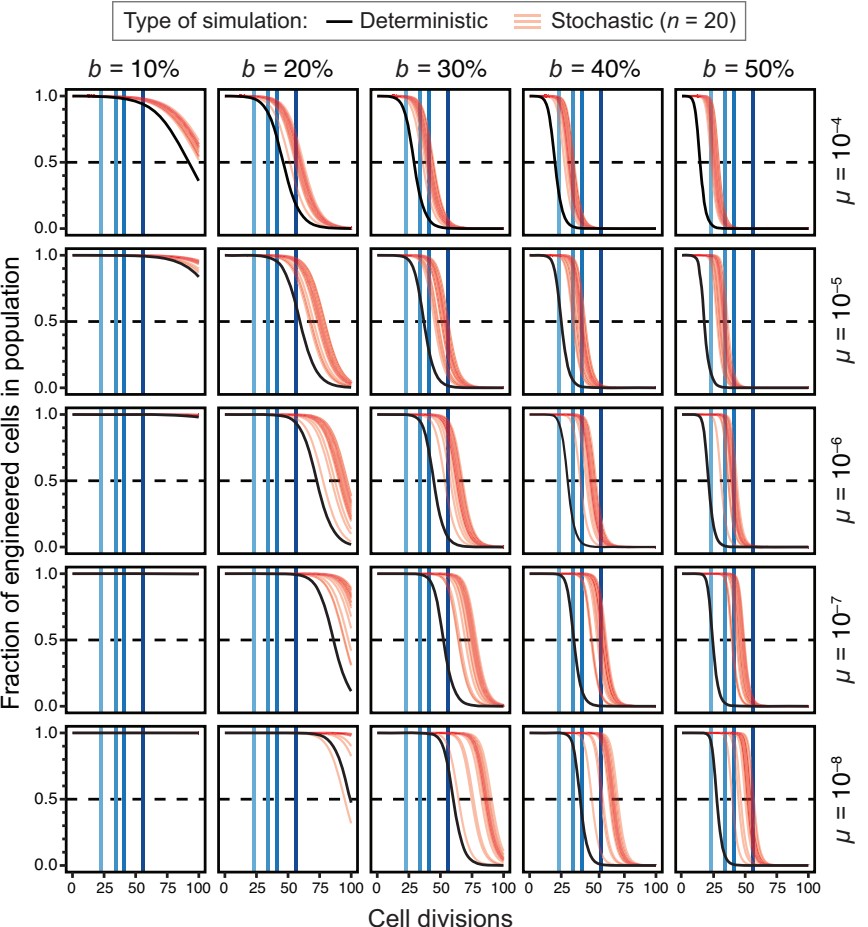

**Fig. 2 | Simulations of evolutionary failure times for populations of engineered cells.** In each panel, the results for deterministic (black) and stochastic (red) simulations of the failure model are shown for one combination of burden ($b$) and failure mutation rate ($\mu$) parameters. Vertical blue lines represent the culture scales shown in Fig. 1C. Curves for stochastic simulations are partially transparent so that one appears pink and overlapping trajectories from multiple simulations appear red. Twenty stochastic simulations are displayed in each panel.

BioBrick part is insulated from elements in the plasmid backbone. pSB1A2 has a transcriptional terminator upstream of the BioBrick prefix multiple cloning site. pSB1C3 has a terminator at the same site and an additional terminator downstream of the BioBrick suffix multiple cloning site.

We measured the growth rates of *E. coli* DH10B derived cells transformed with BioBrick plasmids to determine how many of these genetic parts and devices were burdensome and to what extent (Supplementary Fig. 1). In each microplate assay, we included 5 pSB1C3-based BioBrick plasmids we constructed with different promoter and ribosome-binding site combinations driving expression of blue fluorescent protein (BFP). These plasmids cause different amounts of burden and served as internal controls. We normalized growth rates between assays to account for plate-to-plate variation based on results for the BFP controls and an additional assumption that most parts in each microplate would exhibit no burden (Supplementary Fig. 2, Supplementary Data 2, and "Methods" section).

In total, we measured the effects of the 5 BFP control plasmids and 301 other BioBricks on *E. coli* growth (Fig. 4B, Supplementary Data 3). Of the 301 BioBricks we characterized, we tested 249 in pSB1C3, 40 in pSB1A2, 9 in both of these plasmid backbones, and 3 housed in other backbones (pSB1AK3 or pSB3C5). Even though different antibiotics were added to growth media when testing BioBricks cloned into pSB1C3 and pSB1A2, there was not a significant effect of the plasmid backbone on the growth rates measured for the 9 parts tested in both plasmids ($p = 0.069$, $F_{1,56} = 3.44$, two-way ANOVA) (Supplementary Fig. 3A). We also did not find evidence for any overall difference in the distributions of growth rates measured for parts tested in pSB1C3 versus the other three backbones ($p = 0.92$, two-sided Kolmogorov-Smirnov test) (Supplementary Fig. 3B). Therefore, we considered all of our measurements together, irrespective of the plasmid backbone in which a BioBrick part was tested, in all further analyses.

Excluding the five BFP control plasmids, which were all burdensome, 112 of the 301 other BioBrick part plasmids (37.2% of those tested) significantly decreased *E. coli* growth rates relative to the majority of parts that had no burden before correcting for multiple testing (individual one-tailed $t$-tests, $p < 0.05$). For 31 BioBricks the growth rate burden was significantly greater than 10%, for 19 it was significantly greater than 20%, and for 6 it was significantly greater than 30% (one-tailed $t$-tests, $p < 0.05$). In agreement with our population genetic model, none of the BioBrick plasmids had a large enough burden (>45%) that they would be predicted to mutate when growing a small test-tube culture in the laboratory (one-tailed $t$-tests, $p < 0.05$). After accounting for multiple testing using the Benjamini–Hochberg procedure[46] at a 5% false discovery rate (FDR), we can conclude that 59 of the 301 tested BioBrick parts (19.6%) exhibit some level of burden with high confidence (one-tailed $t$-tests, adjusted $p < 0.05$). Table 1 lists the 34 BioBricks that met this criterion and had a mean estimated burden of >10%.

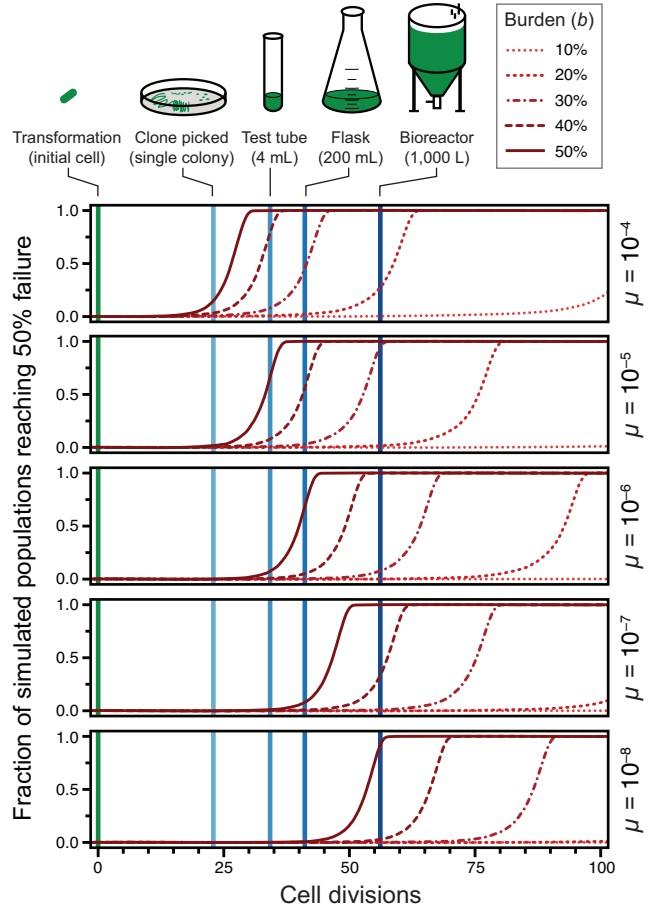

**Fig. 3 | Cumulative distributions of times to 50% failure in stochastic simulations.** Each curve shows the values of this summary statistic of failure for 10,000 simulations with a given parameter combination. More variability in the time to 50% failure leads to a flatter curve.

## BioBricks containing gene expression parts are more likely to be burdensome

Only BioBricks that express an RNA or protein product are expected to appreciably burden a host cell, as the cost of replicating plasmid DNA is generally negligible in comparison[47]. Therefore, we hypothesized that the 59 BioBricks in the high-confidence burden set would be more likely than BioBricks that were not burdensome to contain strong gene expression signals. Series of constitutive promoter parts (J23100–J23119) and ribosome-binding site (RBS) parts (B0030, B0032, and B0034) with known relative strengths are commonly reused in different BioBricks. These promoters and RBS sequences can be divided into weak, medium, and strong variants on the basis of experimental data reported in the iGEM Registry (Fig. 5A, B)[48].

We examined whether BioBricks that exhibited burden were more likely to include these common gene expression parts than those that were not burdensome (Fig. 5C, D). BioBricks that contained any of these constitutive promoters were 2.9 times as likely to be in the set of 59 burdensome BioBricks compared to those that did not have one of these promoters ($p = 0.00040$, Fisher's exact test), with a trend that the stronger promoters were even more likely to be associated with burdensome BioBricks. Similarly, BioBricks that included the strongest of the three RBS parts (B0034) were 2.1 times as likely to slow *E. coli* growth as BioBricks that included only the two weaker RBS variants or none of the RBS sequences in this series ($p = 0.0037$, Fisher's exact test). None of the BioBricks that contained the medium-strength RBS also had a constitutive promoter part, which can explain why this

category noticeably deviated from the general trends. Overall, these results agree with the general expectation that strong, constitutive gene expression contributes to the burden of many BioBricks.

One case that stood out in examining these results was BioBrick K880005. It includes the strongest constitutive promoter (J23100) and RBS (B0034) from these sets, but it does not include a downstream open reading frame. Nevertheless, K880005 is among the most burdensome BioBricks that we measured: it reduces the growth rate of *E. coli* by $27.5 \pm 8.6\%$ (95% confidence interval) (Table 1). The high burden of this BioBrick may put it at risk of mutating during laboratory propagation, even at the test-tube scale (Fig. 3). Its unexpected burden could result from transcription and/or translation of sequences downstream of the part in the BioBrick suffix sequence and plasmid backbone. Even though it was tested in the pSB1C3 backbone that has flanking rho-independent terminators designed to insulate the Bio-Brick, there is commonly some level of transcriptional read-through of these elements[49,50].

## BioBrick burden is not correlated with organism of origin

Next, we tested whether BioBricks that incorporated genetic part sequences from certain types of organisms were more likely to exhibit burden. We classified BioBricks according to the most divergent organism from which their sequences originated and examined twelve classification schemes in which we categorized similar organisms at different levels of taxonomic resolution, only considered protein-coding features, and/or omitted fluorescent proteins when assigning a BioBricks's organism of origin ("Methods" section, Supplementary Data 3, Supplementary Fig. 4). We found no variation in the distributions of normalized growth rates of *E. coli* strains carrying BioBrick plasmids with sequences derived from different types of organisms for any of the classification schemes (Kruskal–Wallis tests, $p > 0.5$) (Supplementary Table 1). The chances that a BioBrick was classified in the high-confidence burden set of 59 BioBricks also did not vary significantly with source organism category for eleven of the twelve classification schemes (likelihood-ratio tests comparing binomial models, $p > 0.05$) (Supplementary Table 1). The one exception, which only barely reached statistical significance ($p = 0.047$), found that BioBricks with sequences from bacteria other than γ-proteobacteria were more likely to be burdensome compared to sequences from *E. coli*, γ-proteobacteria other than *E. coli*, or eukaryotes (Supplementary Fig. 4B). Overall, we conclude that organism of origin did not consistently correlate with whether a BioBrick would exhibit burden or the amount of that burden.

## Mutations and variability in strains with BioBrick plasmids support a burden limit on constructability

To validate the identity and integrity of the plasmids we tested, we compared whole-plasmid sequencing data for 215 BioBricks plus the 5 BFP controls to the sequences reported in the iGEM Registry (Supplementary Data 4 and "Methods" section). Excluding the controls, we sequenced 214 of the 301 BioBricks for which we had burden measurements (71.1%). Of these, 8 plasmids were initially misassigned to the wrong BioBrick and 3 others to the wrong backbone in our results before we corrected them. For 185 of the 215 sequenced plasmids (86.0%), our results perfectly matched the expected BioBrick sequences. Of the 30 others, we found relatively minor discrepancies between the sequencing data and the reported BioBrick sequences for 23, and the other 7 had major discrepancies, such as large deletions or transposon insertions.

It is not possible to determine with 100% certainty whether these discrepancies are due to errors in the designed part sequences that were submitted to the iGEM Registry or mutations that arose and took over cell populations because they reduced the BioBrick burden. Most discrepancies are single base changes or deletions that may have no effect on genetic part function. However, in the seven cases of major

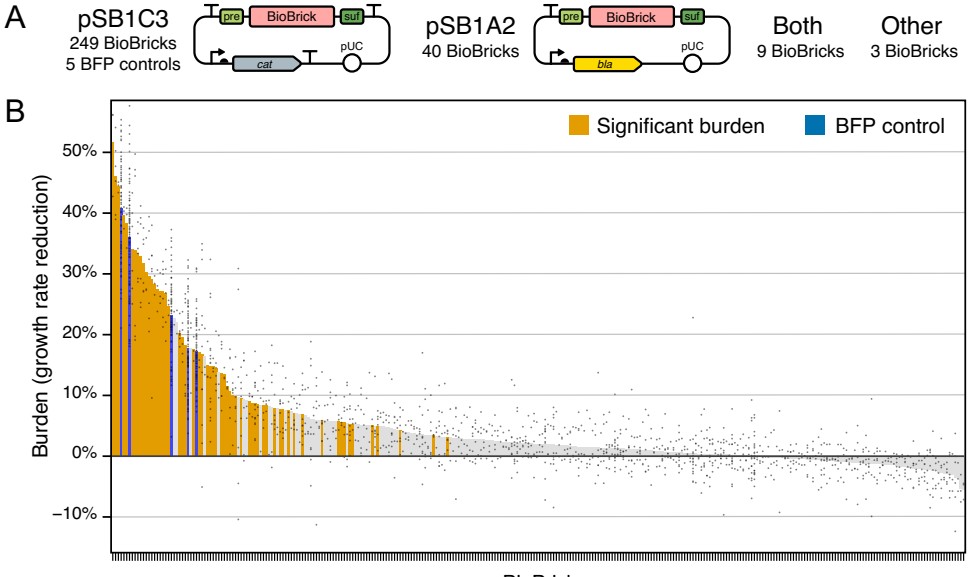

**Fig. 4 | Measurements of BioBrick burden. A** Maps of the two plasmid backbones that housed most of the 301 BioBricks that were tested and the five BFP controls that were included in every assay. The prefix (pre) and suffix (suf) multiple cloning sites used in BioBrick assembly are shown. **B** Burden of each BioBrick tested. Burden is the percentage reduction in the growth rate of *E. coli* cells transformed with a BioBrick plasmid. Gray points are individual measurements. Bars are the means for all measurements of a BioBrick. For BioBricks with orange bars, the measured burden was significantly greater than zero (adjusted $p < 0.05$, one-tailed $t$-tests with Benjamini–Hochberg correction for multiple testing). Supplementary Data 2 contains the results of all microplate assays. Supplementary Data 3 has calculated statistics. Source data are provided with this paper.

discrepancies, we can be reasonably sure that we have observed unplanned mutations with consequences. Two BioBricks (S03749, I759016) were inactivated by insertion sequence (IS) elements that must have transposed into their sequences after construction. Two BioBricks that were closely related to the second of these (I759019, I759020) had frameshifting or large deletions. Two other parts related to one another (K523020, K523022) also contained large deletions, and the first of these was marked as "believed to contain major errors" in the iGEM Registry. Finally, most of BioBrick I732920 was deleted, and its sequence marked as "inconsistent" in the iGEM Registry.

Two of the BFP control BioBrick plasmids, which our own iGEM team constructed and submitted to the iGEM Registry, demonstrate that there is a real risk of selecting cells that have mutated copies of highly burdensome plasmids soon after they are created. We noticed that there was a discrepancy in the order of the growth rates of strains carrying these plasmids in our burden assays: the two control plasmids designed to have the strongest combinations of promoters and ribosome-binding sites driving BFP expression unexpectedly exhibited the least burden. Re-testing the frozen cell stocks of the original transformants of these plasmids demonstrated that the derived stocks used in the burden assays had picked up mutations that largely alleviated the burden of these two plasmids (Supplementary Fig. 5). The burden was reduced from 45.8% to 17.8% in one case and from 41.9% to 17.2% in the other. Further supporting the instability of the two most burdensome BFP control plasmids, when we shared them with another iGEM team, they found an insertion of an IS5 element occurred in the promoter driving BFP expression in their transformant, which reduced but did not eliminate fluorescence.

Even if the original cell giving rise to a colony that is picked after transforming a plasmid or restreaking from a stock has only intact copies of a plasmid, it may give rise to a heterogeneous population of descendant cells as it is cultured, stored as a frozen stock, and revived. As our simulations show, more burdensome plasmids will be at a greater risk of having newly evolved mutants begin to take over the population during these steps. If this type of stochastic, partial

takeover of a cell population with mutants was occurring during our experiments, more burdensome BioBricks might exhibit greater variability in their measured growth rates between replicate cultures. In agreement with this hypothesis, we found a significant trend toward a higher standard error of the mean for growth rates measured for BioBrick plasmids that had higher burden ($p = 2.0 \times 10^{-11}$, two-tailed $t$-test for a non-zero slope) (Supplementary Fig. 6).

In summary, two lines of evidence support that clonability or constructability limits for engineered DNA are creating an upper bound on what plasmids are possible to construct and measure that might be causing us to underestimate the burden of some BioBrick designs. First, our BFP control plasmids designed to have the strongest gene expression mutated during construction, and some of the BioBrick plasmids we characterized also sustained mutations that likely reduce their burden. Second, we see more variation in our measurements of growth rates for more burdensome BioBricks, which could be at least partially explained by cells with mutations that reduce plasmid burden arising and beginning to take over during our assays.

**Redirecting gene expression capacity to recombinant protein production causes a proportional reduction in growth rate**
The *E. coli* DH10B-GEM strain that we used as a host for testing BioBrick burden has a constitutively expressed GFP gene integrated into its chromosome (Fig. 6A). This GFP can be used to monitor how much the presence of a BioBrick plasmid reduces the capacity of an *E. coli* cell for expressing its native proteins[21,51]. If the main source of burden from a plasmid is due to its use of any cellular resources or machinery that are necessary to achieve translation of proteins (e.g., ribosomes), then one expects that for a given reduction in GFP expression, there will be a proportional reduction in growth rate. If there is a reduction in growth rate that is larger than expected relative to the reduction in GFP expression that is observed, then some or all of the burden comes from other sources. For example, gene products encoded on the plasmid may lead to depleting a cellular resource that is not directly related to gene expression or have a toxic effect that interferes with homeostasis.

## Table 1 | Most burdensome BioBricks

| BioBrick[a] | Seq[b] | Burden (b)[c] | Fraction other burden (b_O/b)[d] | Subparts[e] | Function[f] |
|---|---|---|---|---|---|
| K523022 | M | 51.7 ± 19.2% | n.s. | $P_{lac}$ &lacZ' crtE crtI crtB | Carotenoid synthesis (Pantoea ananatis) |
| K733010 | C | 46.0 ± 6.2% | 0.16–0.71 | $P_{tms}$ &endB | Antitoxin gene (Bacillus subtilis)[B] |
| J04450 | NS | 44.4 ± 2.2% | n.s. | $P_{lac}$ &mRFP1 | RFP reporter |
| K523014 | C | 39.6 ± 4.7% | 1.04–1.98 | $P_{lac}$ &lacZ' bglX | Cellobiose degradation |
| K523020 | M, E | 38.3 ± 8.7% | n.s. | $P_{lac}$ &lacZ' INP+bglX | Cellobiose degradation (INP, Pseudomonas syringae) |
| K608010 | C | 34.1 ± 7.7% | NT | $P_{J23110}$ &GFP | GFP reporter |
| K515100 | C | 33.9 ± 15.9% | 0.26–0.88 | $P_{veg2}$ &IaaM &IaaH | Indoleacetamide synthesis (Pseudomonas savastanoi)[B] |
| J61000 | m | 33.4 ± 4.1% | 0.21–0.96 | $P_{cat}$ &cat | Chloramphenicol resistance |
| K541526 | C | 32.9 ± 7.8% | n.s. | $P_{veg}$ &reflectin1A | Reflectin reporter (Euprymna scolopes)[B] |
| K592020 | m | 31.8 ± 5.0% | NT | $P_{fixK2}$ &cl(λ) $P_{cl}$ &amilCP | Blue light sensor output (Acropora millepora) |
| J36335 | m | 30.2 ± 12.2% | n.s. | $P_{lac}$ &kaiA $P_{lac}$ &kaiC | Circadian rhythm (Synechococcus elongatus) |
| I759017 | C | 29.5 ± 8.3% | NT | $P_{tet}$ [cis5] &YFP | YFP reporter |
| K346000 | C | 29.1 ± 10.4% | n.s. | &RNAP(T3) | Phage RNA polymerase (Phage T3) |
| C0056 | C | 28.2 ± 3.9% | n.s. | cl434(λ) | Mutant phage repressor (Phage λ) |
| K880005 | C | 27.5 ± 8.6% | n.s. | $P_{J23100}$ & | Gene expression |
| C0053 | NS | 27.2 ± 6.5% | n.s. | cII(P22) | Phage repressor (Phage P22) |
| K608012 | C | 27.1 ± 4.7% | NT | $P_{J23110}$ &GFP | GFP reporter |
| I759014 | C | 26.8 ± 5.8% | n.s. | $P_{tet}$ [cis2] &YFP | YFP reporter |
| K541502 | C | 24.6 ± 3.2% | 0.42–1.91 | $P_{veg}$ &lipA_{sig} | Gene expression/secretion (Bacillus subtilis)[B] |
| K395602 | C | 20.3 ± 1.9% | 0.09–0.38 | $P_{T7}$ &MpAAT1 | Apple fragrance generator (Malus pumila) |
| K733013 | C | 19.5 ± 3.3% | n.s. | $P_{veg}$ &GFP | GFP reporter[B] |
| K523013 | C | 18.3 ± 8.8% | NT | $P_{lac}$ &lacZ' INP + EYFP | EYFP reporter (INP, Pseudomonas syringae) |
| I761014 | C | 17.5 ± 5.0% | 0.21–1.33 | &cinR &cinI | Quorum sensing (Rhizobium leguminosarum) |
| C0051 | NS | 17.1 ± 8.2% | n.s. | λ-cl + LVA | Phage repressor (Phage λ) |
| K137018 | C | 16.8 ± 8.2% | NT | $P_{L-lacO1}$ &luxR $P_{lux-R}$ &GFP | Quorum sensing receiver (Aliivibrio fischeri) |
| K1149051 | C | 15.0 ± 8.4% | n.s. | $P_{J23104}$ &phaC1 phaA phaB1 | Polyhydroxybutyrate synthesis (Ralstonia eutropha) |
| K731721 | C | 14.8 ± 4.4% | n.s. | | Transcription terminator (Phage T7) |
| K639003 | m | 14.8 ± 2.8% | n.s. | $P_{rrnB-P1}$ &lacI $P_{L-lacO1}$ &mCherry | Stress sensor |
| K541501 | C | 14.4 ± 3.6% | n.s. | $P_{veg}$ &sacB_{sig} | Gene expression/secretion (Bacillus subtilis)[B] |
| K608011 | C | 13.7 ± 5.4% | NT | $P_{J23110}$ &GFP | GFP reporter |
| K861172 | NS | 13.4 ± 2.5% | n.s. | $P_{cstA}$ &cl(λ) | Phage repressor (Phage λ) |
| K617004 | C | 11.6 ± 1.5% | 0.95–2.32 | attP(λ) P'OP | Phage attachment site (Phage λ) |
| K325218 | m | 10.8 ± 7.3% | 0.76–1.55 | $P_{araC}$ &luc(orange) | Luciferase reporter (Luciola cruciata) |
| I712669 | m | 10.1 ± 4.5% | NT | $P_{CMV}$ GFP | GFP reporter[M] |

[a]BioBrick accession numbers. The 34 parts shown all had an estimated burden that was significantly greater than zero after correcting for multiple testing (one-tailed t-tests, Benjamini–Hochberg adjusted $p < 0.05$) and had a mean estimated burden value of >10%.

[b]Results of sequencing the BioBrick plasmid: C, reported BioBrick sequence was confirmed; M major discrepancies found in BioBrick sequence; m minor discrepancies found in BioBrick sequence; NS not sequenced; E part is reported to have errors in the iGEM Registry. Full sequencing results are provided in Supplementary Data 4.

[c]Burden as the percentage reduction in growth rate caused by the BioBrick ± estimated 95% confidence limits.

[d]95% confidence interval on the fraction of burden from sources other than utilization of the host cell's gene expression capacity. n.s., value was not significantly greater than zero (one-tailed t-tests, Benjamini–Hochberg adjusted $p < 0.05$). NT not tested because the BioBrick contains a protein that interferes with the measurement of GFP fluorescence.

[e]Representation of gene expression signals and genes in the BioBrick abbreviated as follows: $P_x$, promoter from gene or operon x; &, ribosome-binding site; [y] other regulatory sequence. Other italicized entries are gene names.

[f]General description of the designed function of the BioBrick. For BioBricks that contain recombinant DNA encoding genes other than fluorescent proteins, the organism of origin is shown in parentheses. Superscript B or M, indicates that the gene expression sequences are intended to function in Bacillus subtilis or mammalian cells, respectively.

To establish that the monitoring device worked as expected, we initially tested two series of plasmids that express other fluorescent proteins (FPs) at varying levels (Fig. 6B). The first was our set of 5 burdensome BFP control plasmids that have different promoter and RBS combinations. Here we used stocks of cells with the BFP plasmids that did not contain the mutations that alleviated the burden noted above. The second set consisted of 14 plasmids available from the iGEM Registry that contain constitutive promoters of different strengths driving the expression of RFP. These RFP constructs were not included in the prior tests of BioBrick burden because they are housed in a different plasmid backbone (J61002). In both cases, we expected that all of the burden exhibited by these plasmids would be due to recombinant FP expression depleting the translational capacity of the host cell. FP production does not use any other types of limiting cellular resources, and these FPs are not expected to be toxic to cells within the range of concentrations at which they are expressed.

In agreement with this expectation, we found that the growth rates of these strains were reduced in proportion to how much they reduced GFP expression (Fig. 6C, Supplementary Fig. 7, Supplementary Data 5, Supplementary Data 6). The Pearson correlation coefficients for this linear relationship were 0.93 and 0.81 for the BFP and RFP plasmid series, respectively. The relationship between growth rate and GFP expression differed slightly between the BFP and RFP series, but this was expected because they have different plasmid backbones and were

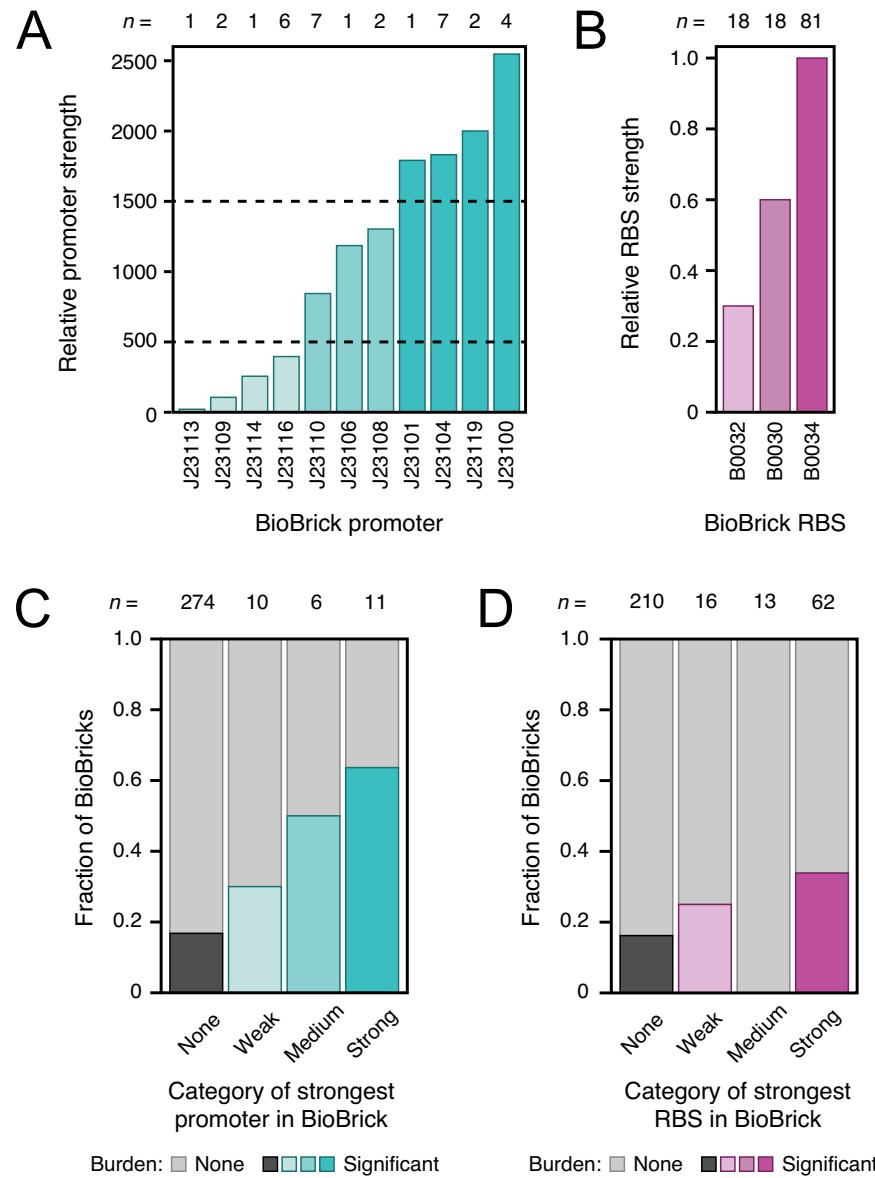

**Fig. 5 | Strong promoter and ribosome-binding sites are more likely to be found in burdensome BioBricks. A**, **B** Relative strengths of common promoters and ribosome-binding site (RBS) BioBrick parts, as reported in the iGEM Registry. The number of examples of each promoter or RBS in the 301 BioBricks examined in this study are indicated above the bars (*n*). Some of these BioBricks contain multiple instances of these promoter and RBS parts. Dashed lines in (**A**) are the thresholds used to classify promoters as weak, medium, or strong. **C**, **D** Fraction of BioBricks that exhibited significant burden (one-tailed *t*-tests, Benjamini−Hochberg adjusted $p < 0.05$) when grouped by the strongest gene expression element of each type that they contain. The total numbers of parts in each category are shown above the bars (*n*). Source data are provided with this paper.

tested under different culture conditions (see "Methods" section). The growth rate reductions seen for RFP series plasmids were roughly in proportion to the amount of recombinant protein that they expressed. By contrast, strains with BFP series plasmids that experienced more gene expression burden did not necessarily produce more BFP. This discrepancy is likely related to how different combinations of promoter and RBS strengths can lead to translating the same amount of protein but with more or less efficient use of ribosomes[21]. As for the 301 BioBricks we tested and the unmutated BFP controls, none of the RFP expression constructs had a burden of >45% in the unclonable range.

**Some BioBricks exhibit burden from sources other than gene expression**

All of our measurements of BioBrick burden were conducted in the *E. coli* DH10B-GEM host strain that contained the GFP gene expression capacity monitor (Fig. 6A), so we next examined how GFP production correlated with the previously characterized growth rates to understand whether the burden of each BioBrick could be attributed partly or wholly to its use of the host cell's gene expression resources. If GFP production was reduced in direct proportion to the growth rate, as it was in the BFP control plasmids, this would indicate that all of the BioBrick burden was from gene expression (Fig. 7A). If there was burden with no or less-than-the-expected reduction in GFP production, then it would indicate a BioBrick was compromising *E. coli* growth for some other reason (Fig. 7B). Of the 301 BioBricks tested, 42 encode GFP or another protein that is expected to interfere with measuring GFP fluorescence, so they were excluded from this analysis (see "Methods" section). We again used the BFP plasmids as internal controls for normalizing GFP production rates between different microplate assays (Supplementary Fig. 8 and "Methods" section).

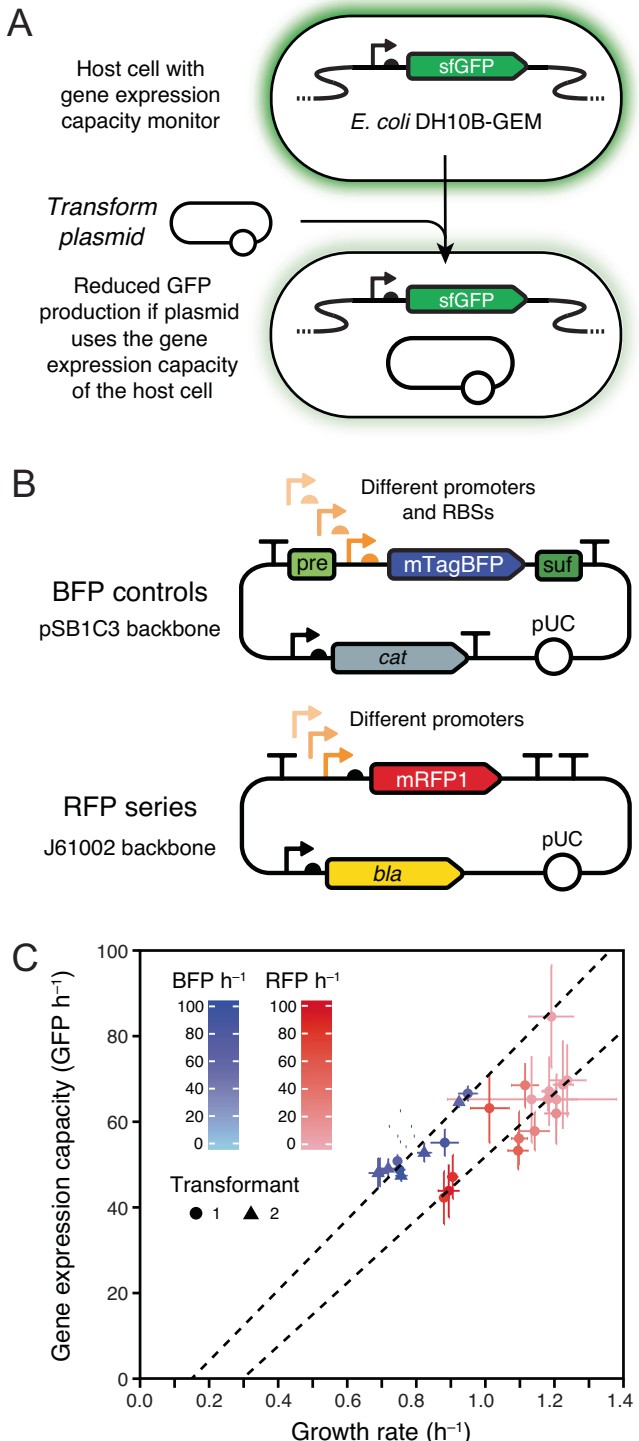

**Fig. 6 | Expression of recombinant proteins from a plasmid reduces the growth rate of *E. coli* because it diverts some of its capacity for gene expression. A** *E. coli* DH10B-GEM host strain with the gene expression capacity monitoring device that constitutively expresses GFP integrated into its chromosome. **B** Maps for the BFP and RFP plasmid series. **C** Growth rates and fluorescent protein production rates for different BFP and RFP plasmids in *E. coli* DH10B-GEM. Dashed lines are Deming regressions showing that the reduction in growth rate is proportional to the reduction in the capacity of the host cell for protein expression within each set of strains. The rate of GFP production from the monitoring device is used as a readout of gene expression capacity. Rates of BFP and RFP production in cells with each type of plasmid are indicated by shading in the respective color. Means are

plotted with error bars that are 95% confidence limits. Two independent transformants of each BFP plasmid that were tested separately are displayed as points with different shapes. There were eight technical replicates (microplate wells) for each BFP plasmid transformant and four for each RFP plasmid tested. GFP and BFP production rates were measured on different relative scales and each series uses a different vector backbone and was measured under different growth conditions, so results should only be compared within each series. Growth rates, GFP production rates, and RFP/BFP production rates for individual plasmids are plotted in Supplementary Fig. 7. Supplementary Data 5 contains the results of all microplate assays, and Supplementary Data 6 has calculated statistics. Source data are provided with this paper.

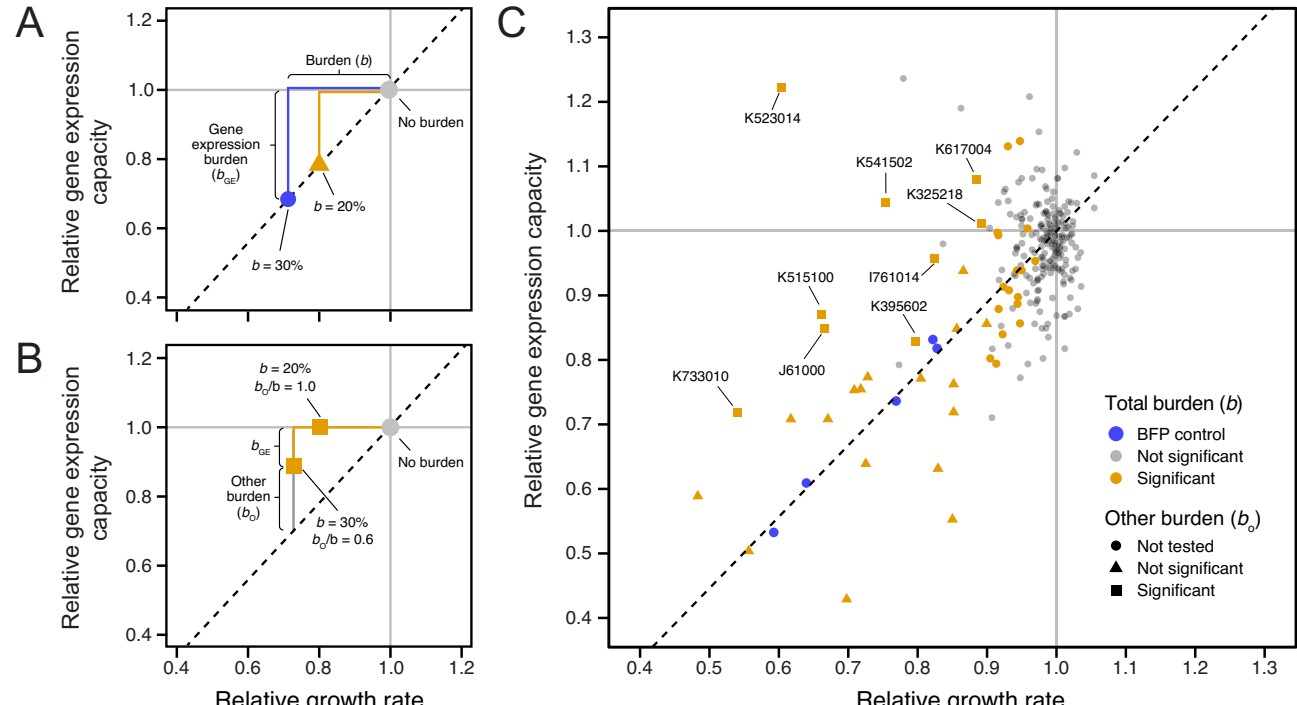

**Fig. 7 | Some BioBricks exhibit burden from sources other than gene expression. A** Examples of expected results for two BioBricks that exhibit burden (*b*) that is wholly due to utilizing the gene expression capacity of the host cell. The reduction in growth rate is proportional to the reduction in GFP production according to a linear relationship (dashed line) that is established from measurements of control strains. **B** Examples of expected results for two BioBricks that exhibit burden from sources other than gene expression. **C** Mean growth rates and GFP production rates measured for 259 BioBricks that do not contain fluorescent proteins that are expected to interfere with measuring GFP fluorescence in the *E. coli* host strain containing the gene expression capacity monitor. Points for each BioBrick are colored based on whether the burden (reduction in growth rate) was significantly greater than zero (one-tailed *t*-tests, Benjamini–Hochberg adjusted $p < 0.05$). Symbols indicate whether the null hypothesis that all burden was due to utilizing the gene expression capacity of the host cell could be rejected (one-tailed *t*-tests, Benjamini–Hochberg adjusted $p < 0.05$). Only the 26 BioBricks with burden significantly greater than zero, a mean estimated burden >10%, and no interference from other fluorescent proteins were tested. BioBricks with significant burden from sources other than gene expression are labeled with their accession numbers. Estimates of $b_O/b$ for these BioBricks are shown in Table 1. Source data are provided with this paper.

Plotting a linear relationship between the BFP plasmid controls, the no-burden BioBrick plasmids, and the origin yields the expected trade-off between growth rate and GFP production for the BFP plasmids and some of the measured BioBrick plasmids (Fig. 7C). However, some parts displayed a higher GFP production rate than what would be expected from the measured growth rate reduction, evidence that some or all of their burden arises for reasons other than diverting the host cell's gene expression resources. Of the 26 BioBrick parts with a high-confidence prediction of burden and a mean estimated burden >10% that could be evaluated in this assay, 9 (34.6%) had a significantly greater reduction in growth rate than predicted from the change in GFP production (adjusted $p < 0.05$, one-tailed *t*-tests with Benjamini–Hochberg correction for multiple testing), indicating that a component of their burden is due to a source other than reducing the gene expression capacity of the host cell (Table 1).

## Discussion

By measuring the burden of 301 BioBricks and performing computational simulations, we established an evolutionary limit on the constructability of engineered DNA sequences: none of the BioBricks we tested slowed *E. coli* growth rates by >45%. Our results are in broad agreement with other studies that have made similar measurements of growth defects and the effects of spontaneous mutations that alleviate the burden of engineered DNA on bacterial cells[8,11]. For example, researchers testing a library of plasmids expressing three fluorescent proteins found that a mutant that deleted one of these genes and took over populations after 30 generations of serial transfer had an 89%

higher exponential growth rate compared to the original engineered strain[10], which corresponds to this mutation reducing burden by 47%. Similarly, the level of burden under non-inducing conditions topped out in the 40–60% range for cells containing various constructs in the study that developed the gene expression capacity monitor we used[21].

We found potential mutations in some BioBricks relative to their designed sequences and more variation in our measurements of more burdensome BioBricks. We also discovered that two of the BioBricks we used as internal controls for our assays unexpectedly mutated while we were using them in ways that maintained some BFP fluorescence yet reduced their burden from near the unclonable threshold (>40%) down to levels that can be reliably maintained during growth on a laboratory scale (<20%). These results suggest that we may be underestimating the burden of some BioBrick designs, either because their plasmids were mutated before we obtained them or because new mutants arose and reached appreciable frequencies in our assays. Some discrepancies are likely due to human errors in the sequences digitally submitted to the Registry versus the original DNA samples themselves. For example, researchers might have copied over a portion of a sequence from a prior plasmid map or part entry and assumed it was correct and unchanged without ever empirically validating their construct. However, there is also both direct and anecdotal evidence that some Biobricks are prone to mutate.

One such example of evolutionary instability is for the exceptionally well-characterized BioBrick F2620 device, which encodes an inducible luciferase gene[29]. While not one of the BioBricks we tested, F2620 was previously noted to reproducibly fail in <100 generations

due to deletions between two 143-bp repeats introduced by re-use of the B0015 double terminator part. The creators confirmed that this failure consistently occurred due to mutations that happened after plasmid transformation. Our model shows how you can get deterministic failures like this if the mutation rate is sufficiently high, as it can be for repeat-mediated deletions[40]. We discovered inactivating deletions or transposon insertions in seven BioBrick plasmids, which likely indicates that they are also especially prone to mutational failure. As an example, the Registry page for BioBrick K523020—one of the most burdensome plasmids that we measured—contains a warning, "Part submitted to Registry is believed to contain major errors," which is probably more typical of how a user of an unstable part would understand rapid evolutionary failure due to mutations that are relieving the burden.

Future work could clarify whether the cases of sequence discrepancies we encountered are already mutated BioBricks or design errors by reverting the putative mutations to the designed sequences and, if successful (i.e., the change does not make them so burdensome that they are unclonable), measuring their burden. Alternatively, deep-sequencing populations of plasmids isolated from laboratory-scale cultures could be used to characterize whether they consist of mixtures of mutated and unmutated plasmids[11,52]. Surveys of plasmids in other repositories have also found that some acquire inactivating transposon insertions[53]. Our findings support calls for researchers to report the full sequences of plasmids they create and submit to repositories such as the iGEM Registry[54,55] and caution that one should also verify the sequences of plasmid stocks obtained from repositories. This information will make it possible to recognize when evolution is undermining DNA constructs and experimental results.

The GFP gene expression monitor that we used responds to changes in a cell's global capacity for protein expression. For any one construct, this could theoretically represent depletion of factors as diverse as the availability of RNA polymerases, ribosomes, initiation factors, charged tRNAs, amino acids, or nucleotides. However, we expect that ribosome availability is the limiting factor in all or nearly all BioBricks we tested, based on studies of recombinant protein overexpression in *E. coli*[14–18]. While we were able to establish overall trends that plasmids containing strong constitutive promoters and ribosome-binding sites had a higher chance of exhibiting burden, it was not possible to predict the gene expression component of burden a priori on this set of sequences. With the limited set of BioBricks we tested, we were also unable to examine whether the burden of individual genetic parts can be used to predict the burden of complex devices constructed from combinations of these parts. Ongoing improvements in tools for predicting transcription and translation initiation rates and expanding databases of high-throughput measurements[56,57] may eventually make these types of predictions possible. Our finding that the type of organism from which a BioBrick's sequence originated was not consistently correlated with its burden agrees with a high-throughput study of horizontal gene transfer from 79 prokaryotic genomes to *E. coli*[58]. It concluded that differences in gene expression were more important than the source organism's relatedness to *E. coli* in explaining why plasmids containing certain genes could not be cloned in this host.

Burden can also arise for diverse reasons other than gene expression, anytime engineered DNA taxes a cellular resource to the extent that it becomes a bottleneck for cell growth. For example, genetic engineering can overwhelm protein export pathways or the capacities of different subcellular compartments[25,26]. Further case studies of the burdensome plasmids with costs not associated with gene expression could reveal the origins of these costs. It would be particularly useful to create other types of burden monitors, e.g., of protein secretion, membrane occupancy[59], or different metabolic bottlenecks so that the relevant limiting factors could be rapidly diagnosed and systems redesigned accordingly to make them more

stable. This more refined information will likely be needed to predict how the burden of a composite part or device depends on the burden of each of the genetic parts from which it is constructed. If multiple components use gene expression resources, then one might expect them to have additive effects on the burden, but if they use orthogonal (i.e., distinct) limiting resources, then one may find that the combination is no more burdensome than the more burdensome of the two on its own.

We measured burden as a decrease in the exponential growth rate of *E. coli* host cells. While this was convenient for making replicated, high-throughput measurements in a microplate reader, it does not fully reflect how a DNA construct impacts the evolutionary fitness of a cell. For example, it is possible that engineering a cell changes the lag time before growth begins[60], survival during stationary phase, colony growth on agar, or survival of cryopreservation. Furthermore, our approach can only be applied to understand genetic stability under laboratory conditions, not in environmental contexts or host-associated microbiomes. Co-culture competition assays between a strain of interest and a reference strain could be used to measure fitness in a way that captures all components of fitness in any environment[61]. To make these measurements high-throughput, host strains with unique sequence barcodes in their chromosomes and transformed with different engineered plasmids or DNA constructs could be simultaneously competed all-against-one-another in bulk competitive fitness assays[62,63]. Our experiments were all in a cloning strain of *E. coli*. It would be interesting to examine how burden varies in strains optimized for other applications, such as recombinant protein production.

Researchers can take actions to improve the constructability and stability of especially burdensome engineered DNA sequences. Most obviously, using low- or medium-copy plasmids rather than high-copy ones or integrating single-copy constructs into the chromosome of a bacterium will often reduce the burden into the cloneable and stable ranges[10]. Systems have also been engineered for controlling plasmid copy number, so that DNA parts can be maintained in cells at a low copy number and then amplified on demand[47,64]. Similarly, reducing the burden of a construct can be achieved by altering promoter and ribosome-binding site strengths or by using inducible promoters, as long as these changes are compatible with device function[10,21]. Systems that regulate expression in response to the growth rate of a cell[65,66] or that couple continued functioning of the engineered DNA to cell survival[67] can more directly buffer against evolutionary failure. Another category of more ambitious approaches is to introduce orthogonal polymerases[68] or ribosomes[69,70] into a cell to prevent synthetic constructs from competing with native gene expression, though the requirement that a cell produce the necessary machinery may itself be burdensome. Next, aspects of the growth environment can sometimes be changed. For example, supplementing media with vitamins or altering salt concentrations has been reported to stabilize certain constructs[11,22]. A final category of approaches seeks to reduce the chances of mutations to improve the evolutionary stability of genetic constructs[7,71]. For example, cells with lower mutation rates can be created by deleting or repressing transposons[9,72] or by altering cellular processes that affect point mutation rates[12,73].

We created an interactive model of failure mutations in a cell population that can be used to explore how tuning mutation rates and construct burden affect whether a DNA construct is likely to remain intact within cell populations that are grown to typical laboratory and production scales. Similar deterministic[6,11] and stochastic[74,75] models have been developed by others. Models that include individual steps in gene expression and RNA and protein degradation are also beginning to be used to examine evolutionary stability[21,76]. Our model and these others still do not consider or fully take into account several complications[71]. First, rather than one category of mutation leading to complete failure, there are typically multiple categories of mutations,

some of which only partially alleviate the burden, occurring at different rates in real systems[10,11]. Second, burden, plasmid copy number, and mutation rates are not necessarily independent parameters and may exhibit cell-to-cell variation. For example, overexpression of recombinant proteins can activate stress responses and increase mutation rates[73]. Third, plasmids are multicopy within cells so the fitness benefit of a mutation can take several generations to fully manifest and depends on how plasmids segregate between daughter cells. These intricacies of plasmid evolution have been tackled by a variety of more complex models that could be applied to engineered plasmids[77]. Finally, models that take into account different phases of cellular growth could be used to further refine these dynamics[78].

Modeling cellular function over time in a way that takes evolution into account is important for making synthetic biology more reliable and predictable. This random, self-reinforcing failure mode does not have a direct parallel in traditional engineering fields. Researchers designing engineered cells should be aware of when they are nearing a danger zone of evolutionary stability where DNA designs may become unconstructable. They should also recognize that the stochastic nature of evolutionary failure may lead to large variation in their experimental results, failure during process scale-up, or loss of function when cells are deployed for long periods of time in complex environments outside of the lab, such as in animal and plant microbiomes. Our interactive model (https://barricklab.org/burden-model) can be used for educating both new and practicing synthetic biologists about evolutionary constraints. In the long term, it is critical that we improve our understanding of what synthetic DNA constructs exhibit burden and why. Our results from measuring the burden of many BioBricks contribute to this goal. The overarching conclusion of our study can be summarized as a rule of thumb: to avoid the specter of unwanted evolution, don't attempt to engineer a microbial cell in a way that slows its growth rate by >30%.

## Methods

### Model of evolutionary failure

We implemented a model in R that is similar to one used by Rugbjerg et al. to predict loss of production from an engineered cell population due to escape mutations[11]. We parameterized our model such that failed (i.e., mutated) cells, $F$, have a relative growth rate of one. Engineered cells, $E$, have a growth rate that is this value minus the burden, $b$, of the engineered construct. The corresponding equations for how the numbers of engineered cells, $E(t)$, and failed cells, $F(t)$, change over time are:

$$\frac{dE(t)}{dt} = (1 - b)E(t) - \mu(1 - b)E(t) \quad (1)$$

$$\frac{dF(t)}{dt} = F(t) + \mu(1 - b)E(t) \quad (2)$$

Growth of cells in batch culture typically continues until a certain number of total cell doublings occurs that exhausts the provided resources rather than for a certain fixed period of time. Therefore, we chose to plot the dynamics of engineered and failed cell populations versus the number of cell doublings, $D(t)$, that have occurred at a given time:

$$D(t) = \log_2[E(t) + F(t)] \quad (3)$$

For stochastic simulations of this model, we used the adaptivetau R package[79]. We also created an online version (https://barricklab.org/shiny/burden-model) that can perform deterministic and stochastic simulations of this model using the Shiny R package[80].

## Media and growth conditions

*E. coli* was cultured at 37 °C in Lysogeny Broth (LB) (10 g tryptone, 5 g yeast extract, 10 g NaCl per liter) with 16 g/L agar added for solid media. Unless otherwise indicated, liquid cultures were grown in 18 mm × 150 mm glass test tubes with orbital shaking at 200 r.p.m over a 1-inch diameter. Antibiotics were added at the following concentrations: carbenicillin (100 μg/ml), chloramphenicol (20 μg/ml), kanamycin (50 μg/ml).

## Gene expression monitor strain construction

*E. coli* DH10B-GEM (JEB1203), the host strain used in the burden assays, was created using plasmids and methods described in Haldimann et al.[81] and Ceroni et al.[21]. Briefly, we inserted the constitutive GFP expression cassette cloned into pAH63 (Addgene #66073) into the *E. coli* chromosome at the λ integration site by electroporating this plasmid into DH10B cells containing the helper plasmid pInt-ts (Addgene #66076) and selecting for kanamycin-resistant colonies. pAH63 has a *pir*-dependent R6K origin, so it does not replicate in the recipient cells. pInt-ts has a pSC101ts origin and was cured by screening colonies after further growth at the restrictive temperature of 42 °C to create DH10B-GEM. We also obtained and characterized *E. coli* DH10GFP (Addgene #109392), a strain constructed in the same way in the prior study of burden by Ceroni et al.[21].

We isolated genomic DNA from cultures of DH10B-GEM and DH10GFP using a PureLink Genomic DNA Mini Kit (Invitrogen). Then, we prepared Illumina libraries using 10 μg of DNA as input into a 2 S Turbo DNA Library kit (Swift Biosciences) using 50% reaction volumes and a final PCR step with custom adapters that added dual 6-bp sample barcodes. Sequencing was carried out on a HiSeq X Ten by Psomagen. Reads were compared to *E. coli* DH10B genome (GenBank: NC_010473) and pAH63 plasmid sequences using *breseq*[82,83]. Split-read mappings (new junction evidence) between plasmid and chromosomal sequences verified that the GFP cassette was integrated at the expected site in both strains. There were two shared differences, a single base insertion in an intergenic region and a synonymous base substitution, between both strains and the DH10B reference genome. DH10GFP also had two additional mutations, a nonsynonymous mutation in *uspF* and an IS4 element insertion in *mdtL*.

## Transformation of BioBrick plasmids

We made DH10B-GEM competent cells as follows. A 10 ml liquid culture of cells was grown overnight in a 50 mL Erlenmeyer flask from an aliquot of the glycerol stock. The entire culture was then added to 500 ml of LB in a 2 L Erlenmeyer flask. This culture was incubated until reaching the mid-exponential phase (an OD600 between 0.4 and 0.6). At this point, it was divided into 35 ml aliquots and centrifuged at room temperature for 10 min at 3400 × *g*. Then, the supernatant was removed and all cell pellets were combined by resuspended (via vortexing) in a total of 150 ml of a 10% (v/v) glycerol + 100 mM CaCl₂ solution chilled on ice. Next, 30 ml fractions of the cells were centrifuged again at room temperature for 10 min at 3400 × *g*. Again, the pellets were combined, resuspending in a total of 20 ml of chilled glycerol-CaCl₂ this time. After incubating this mixture on ice for 25 min, 200 μl aliquots were snap-frozen in liquid nitrogen. Competent cells were stored at −80 °C.

Heat shock was used to transform BioBrick plasmids from the iGEM 2018 DNA Distribution Kit into DH10B-GEM. This transformation method entailed transferring 2 μl of a miniprep of the plasmid of interest into 50 μl of competent cells and incubating on ice for 1 h. After this, the mixture was placed in a 42 °C heat bath for 30 seconds and then immediately placed back on ice for another 30 min. Next, we added 950 μl of SOC media and incubated at 37 °C in a shaker incubator for at least an hour. After SOC recovery, we pelleted the cells and decanted 800 μl of the supernatant. We resuspended the pellet in the remaining 200 μl of supernatant and then plated this onto an LB agar

plate with the appropriate antibiotic. After overnight incubation at 37 °C, we picked a colony, grew an overnight culture in liquid LB media, added glycerol to 15% (v/v), and froze a stock at −80 °C.

## BFP plasmid construction

Five control plasmids expressing different levels of mTagBFP were created by assembling BioBrick parts from the iGEM registry. The mTagBFP insert was from part plasmid K592100. It was combined with five promoter + RBS composite parts (K608002, K608003, K608004, K608006, and K608007), by using each of their pSB1C3 part plasmids as the vector backbone in a separate postfixing BioBrick assembly reaction[45,84]. For cloning, we used enzymes from New England Biolabs under standard conditions. Briefly, K592100 was double digested using XbaI and SpeI restriction enzymes in CutSmart buffer. Separately, each of the vector backbones was double digested using SpeI and PstI-HF restriction enzymes in CutSmart buffer followed by incubation with calf intestinal alkaline phosphatase for 1 h. Digested products were then gel extracted and purified using a QIAquick Gel Extraction Kit before being ligated together using T4 DNA ligase. Ligated products were purified using butanol precipitation and then electroporated into competent TOP10 *E. coli* cells. Transformed cells were recovered in SOC for 1 h at 37 °C, followed by plating on LB agar containing chloramphenicol. After incubation at 37 °C for 18 h, we inoculated isolated colonies into fresh LB liquid media containing chloramphenicol and grew these cultures at 37 °C for 18 h. The five resulting composite BioBrick parts were deposited in the iGEM Registry as K3174002, K3174003, K3174004, K3174006, and K3174007.

## Plasmid sequencing

We sequenced BioBrick plasmids isolated from the DH10B-GEM cell stocks that were used for burden assays. In addition, we sequenced plasmids isolated from the TOP10 cell stocks into which the BFP controls were first transformed. Plasmid DNA was purified using a QIAprep Spin Miniprep Kit (QIAGEN) or a PureLink Quick Plasmid Miniprep Kit (Invitrogen). We performed Sanger sequencing on multiple stocks of the BFP control plasmids, in-house Illumina sequencing on these and the other plasmid samples, and outsourced Nanopore sequencing on additional plasmid samples. For Illumina sequencing, up to 10 ng of plasmid DNA was used as input for sequencing library preparation using the 2S Turbo DNA Library kit (Swift Biosciences) with 20% reaction volumes. Custom adapters containing dual 6-bp sample barcodes were incorporated during the final PCR step. The resulting DNA libraries were pooled and sequenced on an iSeq 100 instrument. Nanopore data was obtained from Plasmidsaurus. Porechop[85] and fastp[86] were used to trim adaptors from sequencing reads.

To analyze sequencing results, we first reconstructed the expected BioBrick plasmid sequences from information available on the iGEM Registry webpages (part sequences, vector sequences, and compatibility with different assembly standards). Then, we analyzed Illumina and Nanopore sequencing data in two ways. First, we compared reads to the expected plasmid sequences using *breseq*[82] to see if there were any discrepancies. Second, we performed de novo assembly of reads using either Unicycler[87] or flye[88], annotated the resulting assemblies with pLannotate[89], and examined them for matches to the expected parts using blastn searches[90] against a database of all BioBrick parts included in the 2018 iGEM distribution kit.

## BioBrick plasmid burden assays

We performed burden assays largely as described previously[21]. Strains were revived by adding aliquots of −80 °C freezer stocks to test tubes containing LB with the antibiotic for maintaining their respective BioBrick plasmids. After overnight growth (12–18 h), we vortexed each culture for three seconds and loaded 5 µl into a Nunc MicroWell 96-well optical-bottom plate (ThermoScientific Cat. No. 265301) in triplicate. Every plate included the five control strains (JEB1204-1208), each

also loaded in 5 µl in triplicate, and 12 blank wells (LB only). This arrangement allowed for a total of 23 strains to be tested per plate. To start the assay, a multichannel pipette was used to add 195 µl of LB pre-warmed to 37 °C to every well with pipetting up and down several times to mix. Using a Tecan Infinite Pro M200 Plate Reader, optical density at 600 nm and GFP fluorescence (excitation: 485 nm; emission: 528 nm) were recorded every 10 min with 7 min of orbital shaking during each cycle. Each plate was run for a minimum of 6 h.

## RFP and BFP plasmid burden assays

For the series of plasmids expressing RFP under the control of different promoters, we performed burden assays using the normal procedure plus an additional measurement of RFP fluorescence (excitation: 585 nm; emission: 610 nm). For correlating BFP expression in the control strains to reduced GFP expression, we added a measurement of BFP fluorescence (excitation: 405 nm; emission: 453 nm). The extra fluorescence reads for the RFP and BFP experiments reduced the proportion of shaking time in each measurement cycle, resulting in slower maximum growth rates than were observed with the standard burden assay procedure. RFP samples were measured every 10 min with 6.5 min of shaking during each cycle. BFP samples were measured every 10 min with 7 min of shaking during each cycle. For the RFP series, we also monitored cell density using OD660 instead of OD600 to avoid interference from RFP absorbance[91].

## Burden analysis

To analyze the burden assay data for one plate, we first subtracted the average values of all media blanks from the OD and fluorescence measurements. Next, to deal with well-to-well variation in background levels, we shifted the values to force the means of the points over the first hour of measurements for each strain to match the grand mean for those data points over all replicates of that strain. We then fit growth rates using nonlinear least-squares regression to an exponential model: $C(t) = C_0 \, e^{rt}$. We assumed that OD is directly proportional to the number of cells at a given time, $C(t)$. $C_0$ is the initial number of cells, and $r$ is the specific growth rate. We fit $C_0$ and $r$ for all sets of nine consecutive measurements (a 90-min window in the standard assay) after the OD exceeded 0.03 and recorded the largest value of $r$ as the maximum specific growth rate for that strain. To determine the fluorescent protein (e.g., GFP) production rate per cell, $p$, we repeated this procedure while fitting fluorescence values to the equation: $F(t) = F_0 + C_0 \, (p/r) \, (e^{rt} - 1)$. $F_0$ is the initial fluorescence and $F(t)$ is the fluorescence at time $t$. This equation is derived by integrating the relationship $dF/dt = p \, C(t)$. We fit $F_0$ and $p$ in this model to the data while keeping $C_0$ and $r$ fixed to the values determined from the OD curve fit for the corresponding time window. Again, we recorded the largest value of $p$ across all time points as the maximum fluorescent protein production rate.

To account for plate-to-plate variation in growth and GFP production rate estimates (Supplementary Figs. 2A, and 8A), we normalized measurements made on different plates. In our experimental design, a majority of the plasmids tested in each plate are expected to exhibit negligible burden. This let us estimate the growth and GFP production rates corresponding to 'no-burden' for a given plate by examining the distributions of values measured. Specifically, we calculated the density distributions of growth and GFP production rates using a Gaussian kernel function with bandwidths of 0.014 and 300, respectively, for all non-control strains. To account for multimodal distributions, we took the no-burden value as the highest value among all peaks in the density distribution that were at least 50% as high as the highest peak. Then, we normalized all rate estimates by dividing them by the corresponding no-burden value for that plate (Supplementary Figs. 2B, and 8B). The final distributions of the mean values for each BioBrick plasmid have a major peak at the no-burden value with a noticeable shoulder of strains with a slightly decreased growth rate or

GFP production rate, in addition to some strains with much lower values (Supplementary Figs. 2C, and 8C).

Some BioBricks encode proteins that interfere with measuring GFP fluorescence. Therefore, for the analysis of gene expression capacity and burden, we disregarded all BioBricks described as including GFP; YFP, which has overlapping fluorescence; or the amilCP blue chromoprotein, which strongly absorbs at the wavelength monitored for GFP emission[92]. For the 26 remaining BioBricks that also had growth rate reductions that were statistically significant and mean estimated burdens >10%, we determined whether the observed GFP production rate was compatible with the null hypothesis that all of the burden was due to the BioBrick utilizing the gene expression capacity of the host cells. We determined the expected relationship between growth rate and GFP production rate for purely gene expression burden from measurements of the BFP control plasmids across all plates. Specifically, we used Deming regression to fit this linear relationship, which takes into account measurement errors in both dimensions, and we further required that the fit pass through the no-burden values (i.e., a normalized growth rate of 1.0 and normalized GFP production rate of 1.0). Then, we determined the chance that each BioBrick was located above the BFP regression using a two-dimensional probability distribution of each assuming maximum likelihood $t$-distributions for growth rate and GFP production rate. We took one-half of this value to estimate a one-tailed $p$-value for rejecting the null hypothesis that all of a plasmid's burden was from utilizing the host cell's gene expression resources.

### BioBrick source organism analysis

We classified BioBricks into eight categories depending on the organisms in which their sequences originated (*E. coli*, other Enterobacterales, other γ-proteobacterium, other proteobacterium, other bacterium, fungus, plant, animal, or synthetic) (Supplementary Data 3). BioBricks with multiple parts were classified according to the most divergent category. Virus- and phage-derived sequences were classified with their host species, and codon-optimized constructs were counted with the species in which the amino acid sequence originated. We considered two additional categorization schemes that reduced the number of organism categories to four or two, plus four variants of each of the three categorization schemes in which we did not include features that were not protein-coding genes or did not include fluorescent proteins in assigning the organism of origin to a BioBrick (Supplementary Fig. 4). For all twelve overall classification schemes, we examined whether there were any trends in burden with respect to source organism in two ways. First, we performed a Kruskal–Wallis test to determine whether BioBricks with different categories of source organisms had systematic differences in their normalized growth rates. Second, we determined whether the chances that BioBricks were in the set of 59 burdensome BioBricks differed across organism categories by comparing binomial regression models with and without this factor using likelihood-ratio tests. $P$-values for all twenty-four statistical tests are provided in Supplementary Table 1.

### Statistics & reproducibility

BioBrick plasmids from the iGEM DNA Distribution Kit were transformed based on their locations in these microplates without first examining their sequences or functions. Outlier measurements in the burden assays were identified by manually examining growth curve fits and omitted from the analysis, as documented in the raw data files[93]. No statistical method was used to predetermine the sample size. The experiments were not randomized. The Investigators were not blinded to allocation during experiments and outcome assessment.

### Reporting summary

Further information on research design is available in the Nature Portfolio Reporting Summary linked to this article.

## Data availability

Raw and processed data files and plasmid assemblies are available in a GitHub repository that has been archived in Zenodo (https://doi.org/10.5281/zenodo.11528027)[93]. Raw plasmid and genome sequencing data are available from the NCBI Sequence Read Archive (Accession: PRJNA1090925). Source data are provided with this paper.

## Code availability

Simulation code and analysis scripts are available in the associated GitHub repository[93].

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

## Acknowledgements

We thank Angela Pak, Emily Garcia, Mina Kim, Alex MacAskill, Raul Lopez, and Michelle Chang for performing various experiments and participating in the UT Austin 2019 iGEM team; Daniel Deatherage and Jack Dwenger for assistance with genome and plasmid sequencing; and Giaochau Nguyen, Vrinda Rajkumar, Marco Sanchez, Jeremey Fitzgerald, and Sidharth Kapur from the Microbe Hackers Freshman Research Initiative stream for cloning the BFP control plasmids. We thank the 2019 Michigan State University, Rice University, and Texas Tech University iGEM teams, iGEM judges, and members of the Barrick lab for useful feedback. We acknowledge the Texas Advanced Computing Center (TACC) at The University of Texas at Austin for providing high-performance computing resources. This research was supported by the National Science Foundation (CBET-1554179 to J.E.B., IOS-2103208 to J.E.B., and MCB-2123996 to J.E.B.), the National Science Foundation BEACON Center for the Study of Evolution (DBI-0939454 subcontract to J.E.B.), the National Institutes of Health (R01GM088344 to J.E.B.), and the U.S. Army Research Office (W911NF-20-1-0195 to J.E.B.). The University of Texas at Austin Freshman Research Initiative (FRI) acknowledges support from the Howard Hughes Medical Institute (#52008124). The University of Texas at Austin College of Natural Sciences and Department of Molecular Biosciences provided additional support for the FRI program and iGEM participation.

## Author contributions

Conceptualization: Noor Radde, Genevieve A. Mortensen, Diya Bhat, Shireen Shah, Joseph J. Clements, Sean P. Leonard, Matthew J. McGuffie, Dennis M. Mishler, Jeffrey E. Barrick Data curation: Noor Radde, Genevieve A. Mortensen, Jeffrey E. Barrick Formal analysis: Noor Radde, Genevieve A. Mortensen, Jeffrey E. Barrick Funding acquisition: Dennis M. Mishler, Jeffrey E. Barrick Investigation: Noor Radde, Genevieve A. Mortensen, Diya Bhat, Shireen Shah, Joseph J. Clements, Sean P. Leonard, Matthew J. McGuffie, Jeffrey E. Barrick Project administration: Dennis M. Mishler, Jeffrey E. Barrick Software: Genevieve A. Mortensen, Jeffrey E. Barrick Supervision: Sean P. Leonard, Matthew J. McGuffie, Dennis M. Mishler, Jeffrey E. Barrick Visualization: Noor Radde, Genevieve A. Mortensen, Jeffrey E. Barrick Writing – original draft: Noor Radde, Genevieve A. Mortensen, Jeffrey E. Barrick Writing – review & editing: Noor Radde, Genevieve A. Mortensen, Sean P. Leonard, Dennis M. Mishler, Jeffrey E. Barrick.

## Competing interests

M.J.M. is employed by Plasmidsaurus, which provides plasmid sequencing services for synthetic biology. The remaining authors declare no competing interests.
