## [Peer Review File · Nature Communications]

Reviewers' Comments:

Reviewer #1:

Remarks to the Author:

In this manuscript, the authors propose a simple mathematical model to predict the loss of a desired function within a population of dividing cells as a function of the burden the cell is experiencing and a constant mutation rate. Analysing this model, they were able to verify the design rule – constructs imposing burden of >45% are “unclonable”. The authors then experimentally characterised the burden of ~300 genetic constructs, ranging in complexity from simple parts (promoters, RBSs, etc.) to multi-component pathways. These experiments backed up the theory and showed that burden on shared cellular resources (e.g., ribosomes) was typically the main impact seen.

This was an enjoyable paper to read that aimed to tackle an important problem facing the field. Typically, the issue of burden is discussed in very vague and superficial terms lacking hard numbers. This work directly tackles this issue, providing a valuable contribution that would be of particular interest to: 1. computational biologists requiring data sets to develop new models; 2. researchers in the early stages of their career requiring guidance on basic biological design principles; and 3. the broader bioengineering community by providing insight into underlying mechanisms of burden. The paper was mostly well-written, although grammar could be improved throughout, and several of the sections (specifically the Discussion) could be condensed without hampering the message but improving readability. The figures were clear, and the modelling and experiments appeared to be sound. Access to some of the raw underlying data produced would be beneficial (this did not seem to be present in the initial submission) and would help improve the broader impact of the work. Overall, this paper is a good fit for Nature Communications, but there are some major comments that the authors need to address:

1. The analysis of burden didn't cover the combinatorial nature of the constructs assayed. For example, is the burden of a part additive, multiplicative, or follow some other non-linear relationship when put together with other parts?
2. Figure 5 looks at single components, those too that representative of broad range of construct types characterized in the work. Furthermore, the analysis of the source of burden was rather limited. It would be useful to discuss potential mechanisms of burden beyond promoter and RBS parts, i.e. could it be related to shifts in metabolism, non-codon optimized genes, non-native components (e.g., not from *E. coli* – as many of the BioBricks in Table 1 seem to be)? Some further comments on this point either here or in the Discussion would be helpful.
3. Plasmid copy number can play a major role in burden, especially if they are hosting complex genetic circuits. Furthermore, vectors with a pUC origin of replication (as used here) can have high variability in copy number. It would be interesting to know how far the assumption of similar copy number for plasmids is held across the constructs tested, particularly those imposing the most burden. Testing a subset of constructs exerting different levels of burden with qPCR or some other method could help to address this question and further verify this assumption.
4. The discussion was too long and lacked deeper insights. I would suggest condensing the content where possible.
5. These experiments are all carried out in a cloning strain of *E. coli*. How would they generalize to other strains or organisms? Could the model aid in providing some insight into this question? What effects might other *E. coli* strains have (e.g. MG1655, BL21, MDS24)? Further comments on the applicability of the model beyond the context presented would strengthen the utility of the study.
6. It would have been nice to see some experimental application of this knowledge for forward design of new genetic constructs. This is not essential, but it would raise the impact of the work

beyond observing existing constructs. Alternatively, some discussion on how the information presented could be used to improve design processes would be useful.

I also had several minor comments that the authors should consider:

Abstract: What are the broader implications of this work? It would be nice to state this at the end to help the reader understand the effect this work will have.

Abstract: It may be helpful to include the URL for the online simulation tool.

L41: "in new and more challenging environment" -> "into increasingly challenging environments"

L45: "potentially making their functions unpredictable and unreliable" – perhaps be explicit that evolution can cause cells to modify functions and lead to a breakdown in function?

L54: "constructs must use" -> "constructs use"

L56: Reference 14 is not used correctly, Rouche et al. demonstrated that plasmid copy number does play a role in overall cell burden, especially high copy plasmids.

L59: "synthesize RNAs and proteins" -> "synthesize RNAs and proteins that are not native to the cell"

L59: "Protein overexpression..." – very long sentence that needs to be split.

L64: There are some earlier references to monitoring burden using combined sequencing approaches that are missing: 10.15252/msb.20188719, 10.15252/msb.20167461

L85: "It has been proposed that genetic reliability..." – what do you mean? Could you elaborate a little on what a "certain level" refers to and the issues in defining this property. The allowable fluctuations in function are typically very application dependent so reliability has varying importance.

L86: "sense of for how" -> "sense of how"

L102: "other, currently unknown reasons" -> "other, currently unknown, reasons"

L104: "evolutionary failure can be used to improve the reliability of bioengineering" – is that really what is shown in this work? I would suggest that these large studies are hugely important for understanding the stability of parts and this information *can* be used to aid in the design of more robust synthetic biology.

L110: "alleviate a burden" -> "alleviate the burden"

L112: "will outreplicate and" -> "will replicate more quickly and" – you might also want to use the term "competitive exclusion" which is technically what is happening.

L132: "would be likely to lead to" -> "would likely lead to"

L144: "plays into the mutation rate" -> "affects the mutation rate"

L147: I'm not sure what you mean by "more densely coded". I assume you are alluding to the degeneracy in the genetic code or robustness to mutations. It would be helpful to more explicitly state that.

L153: Please include some other recent examples of mutational hotspots: 10.1038/s41467-021-26286-9

L163 L164, L170: Can you please include the units of the rates you are referring to. Please check throughout and always include the units with any measurement values in the main text.

L177: "these "jackpots" occur" -> "these events occur"

L186: "early jackpots" – I'd suggest not using the term "jackpot". Also, in this context, how can there be a mutation earlier than when the simulation starts. Is it not the fact that multiple mutations are clustered early on, which is unlikely to happen and so rarely seen?

L197: "represent the output" – is that a mean of the simulations? If so, where is the variability?

L208: "We created..." – Some of the discussion in this paragraph would be better suited to the Discussion, not the Results.

L219: "evolutionary constraints predicted" -> "evolutionary dynamics predicted"

L265: "significantly decreased E. coli growth" – I don't understand what you mean by significantly here: statistically significant to the control (may not be very impressive in absolute terms) or seeing a drop greater than some threshold (perhaps more impressive). Could you please be explicit and avoid terms like significant unless explained. In many cases the term "significant" doesn't really add anything and could be omitted and actual changes stated. Please consider this comment throughout the manuscript.

L329: Studies of terminators have shown they are not always perfect and are sometimes highly context dependent. It would be worth mentioning that with some evidence. 10.1038/nmeth.2515, 10.1038/s41467-022-28074-5

L343: Would you not expect to see some mixed colonies? How was the plasmid DNA prepared for sequencing, was clean/reliable data quality seen across entire plasmid sequence? (I believe this information is provided by the full-plasmid sequencing companies)

L355: Does this have implications for how DNA is verified before submission to the registry? This might be nice to add to the Discussion.

L390: "cell for expressing" -> "cell to express"

L446: "reduced in direct proportion" -> "reduced proportionally"

L483: "performing simulations" -> "performing computational simulations"

L497: "their burdens" -> "their burden"

L555: "rate of the E. coli host cells" -> "rate of the host E. coli cells"

L602: "Researchers designing..." – very long sentence that needs to be broken into several.

L610: "do not burden their growth by more than ~30%" – this sentence doesn't make sense, what is burdening growth by a percentage. Perhaps reword to be more explicit, e.g., "ensure your constructs do not lead to more than a 30% drop in normal growth rate"

L683: the authors mention dephosphorylating their DNA digest products using calf intestinal phosphatase, which was followed by gel extraction and ligase with T4-ligase. However,

dephosphorylated DNA cannot be ligated without a phosphorylation step, such as by using T4-poly nucleotide kinase. Is this protocol correct?

L713: Experimental cultures were prepared directly from “-80 °C” stocks – could this affect the results, i.e., starting with a mixed colony?

Figure 1, 2 and 3: It felt like several of these could be combined.

Figure 4: Please provide some representative growth curves for the data presented in Figure 4; this is not necessary for all the constructs, just a small set of samples spanning the range of burden observed to be sure there are no strange growth effects.

Figure 4B and 7C: Both data sets presented in these figures display growth rate for the iGEM constructs and for the BFP plasmids, as well as discuss the “significance” level of the burden imposed on the cells. The colour scheme used to highlight the “significant” burden plasmids is different between the two figures, being yellow-orange and pink for 4B and 7C, respectively. For consistency, it would be helpful for the colours to be the same so that it is easier to draw the link between the two datasets.

Figure 6A and 6B: These could be combined into one panel.

Figure 6C: The relative intensities for the BFP and RFP expression are very difficult to discern from the colour gradient alone. It would be helpful to provide the BFP and RFP fluorescence intensity data on a separate graph, either in the main manuscript or Supplementary Information.

Reviewer #2:

Remarks to the Author:

Reviewer #3:

Remarks to the Author:

Opinion on Radde et al., “Measuring the burden of hundreds of BioBricks defines an evolutionary limit on constructability in synthetic biology”

The authors investigated the the effect of mutation rate and growth retardation, i.e. burden on the stability of genetic parts and devices propagated on plasmids in E. coli. They generated two tunable models, a deterministic and a stochastic to predict the loss of the correct genetic circuit upon culturing. As a rule of thumb, they declare that a burden > 45% leads to unclonability (loss of function already at the stage of colonies obtained after transformation), and recommend to keep the burden below 20-30% if upscaling to 1000 L fermentors is planned.

Next, they measured the burden of 301 BioBricks parts elicited on E. coli, and found 59 to display a >10% burden with high confidence. As expected from the model, no construct displayed a burden >45%. In addition, they found that two of their control plasmids strongly expressing BFP had gone through a mutant selection that reduced their burden from 45.8% to 17.8% and 41.9% to 17.2%, respectively, further supporting that burdens approaching or exceeding 45% will lead to the loss of the correct construct from the bacterial population. The authors found that strong constitutive promoters and strong ribosome binding sites significantly increased the chances of high burden, but specific predictions for the burden of individual constructs could not be made based on this set of data.

Following, the authors used a reporter fluorescent protein, GFP expressed from the chromosome to measure the effect of the cloned genetic construct on the transcription/translation capacity of the host cell. They found that in nearly all cases, the reduction in growth rate corresponded to the reduction of GFP expression, therefore the burden could be attributed to the diverting of the transcription/translation machinery to express the cloned construct. The authors pinpointed a few exceptions, where they provide numerical data for the fraction of the burden caused by other, unknown mechanisms (b_o/b).

The Discussion provides valuable recommendations for the scientific community to reduce the instability of clones, and/or the presence of erroneous plasmids at plasmid repositories. The authors also give good ideas for future research, e.g. the development of further burden monitors, or carrying out high-throughput gene expression measurements to train AI-based predictors of transcription and translation initiation rates.

Overall, the text is reader-friendly, clearly written in perfect English. The selected methods are sound; the analysis of the data applies sufficient statistical analysis. The experiments are well described, and the conclusions are well supported.

Major remark:

The manuscript practically disregards the fact that the burden and the mutation rate are not independent variables. It has been displayed that various stressors, including that of transgene expression elevates the rate of point mutations, deletions and insertions via inducing error-prone polymerase genes, RecA and the transposition of mobile genetic elements, respectively. Even a gene as inert as GFP has been shown to elevate mutation rates 1.5-2 fold, when expressed. I understand that the model was to be kept simple, but this shortcoming could be mentioned in the discussion. The use of engineered bacterial strains displaying reduced mutation rates for cloning purposes is cited. Without overrating these strains, it is worth adding that this phenomenon increases their advantage compared to that inferred from mutation rates measured under unstressed conditions (e.g. *Microbial Cell Factories* 2012, 11:11).

Minor remarks:

Line 180: "deterministic simulations consistently overestimate how unstable a construct will be for a given combination of parameters". Is there evidence that the deterministic models truly overestimate compared to experimental data? If not, I think the authors should add "compared to stochastic models".

Line 387, line 431: the authors refer to a "proportional" reduction of gene expression capacity and the growth rate. I would avoid the word "proportional", unless the fitted line on the correlation plot passes through the origin. On Figure 6, neither of the two lines (especially not the RFP trendline) passes through the origin. The linear correlation is nevertheless evident.

Reviewer #4:

Remarks to the Author:

The authors of this manuscript aim to quantify, both theoretically and empirically, the well-known evolutionary instability challenge in synthetic genetic constructs. The authors first theoretically examine the consequences of the interaction between mutation rate and decreased growth rate, drawing strong and useful quantitative predictions from this simple model. These predictions are then confirmed, leveraging the iGEM parts collection to obtain a large and diverse but highly comparable collection of synthetic constructs, producing specific data, guidance, and protocols that are likely to be of use to many researchers in the future.

Overall, this manuscript was admirably clearly presented and a pleasure to read. I see only a few minor points that need cleanup before its publication:

- p8: It would help to make it clear that the iGEM plasmids are pUC derivatives at this point
- p8: Are you able to numerically estimate mutation rate in the constructs that you measured? It would strengthen things if you could, but I didn't see that in your data.
- Fig 2: use line patterns (e.g., dashes) to make blue lines more distinguishable
- Fig 3: use line patterns or markers to make red lines more distinguishable
- p12: How were the 301 parts chosen? Why do exactly 3 "other plasmid" biobricks? Why the relative balance of pSB1C3 vs. pSB1A2? Was this just all the parts you had available, or was there some other logic behind including some and omitting others?
- p13: Benjamini-Hochberg is likely to be unfamiliar to many readers (including myself): please add a citation, even if it's to a textbook
- p16: The case of K880005 doesn't seem surprising - even if there's no ORF, the promoter and RBS will still be heavily recruiting transcription factors and ribosomes. Moreover, if there are small peptides produced before the promoter, these may be toxic for the cells, as seems to be the case for BCD promoters.
- Fig 6A: it would be good to use the proper SBOL visual symbol for chromosomal integration. See example with the glyph definition at: <https://github.com/SynBioDex/SBOL-visual/tree/master/Glyphs/SequenceFeatures/chromosomal-locus>
- Fig 7C: How do you explain the increased gene expression capacity of K523014 and the gene expression impact higher than growth impact for the few "far below the line" pink triangles? If these are predictable based on construct contents, it would be good to discuss. If these are unexplained variation, then it is not clear whether the other "pink square" points should be considered significant, since there aren't many more of these than there are these anomalies in non-predicted directions.
- p26: typo - "measurement high-throughput"

REVIEWER COMMENTS AND RESPONSES

Please note: all line numbers refer to the **MARKED UP** manuscript version that still contains embedded figures, NOT the main manuscript file that was reformatted for this revision.

Reviewer #1 (Remarks to the Author):

In this manuscript, the authors propose a simple mathematical model to predict the loss of a desired function within a population of dividing cells as a function of the burden the cell is experiencing and a constant mutation rate. Analysing this model, they were able to verify the design rule – constructs imposing burden of >45% are “unclonable”. The authors then experimentally characterised the burden of ~300 genetic constructs, ranging in complexity from simple parts (promoters, RBSs, etc.) to multi-component pathways. These experiments backed up the theory and showed that burden on shared cellular resources (e.g., ribosomes) was typically the main impact seen.

This was an enjoyable paper to read that aimed to tackle an important problem facing the field. Typically, the issue of burden is discussed in very vague and superficial terms lacking hard numbers. This work directly tackles this issue, providing a valuable contribution that would be of particular interest to: 1. computational biologists requiring data sets to develop new models; 2. researchers in the early stages of their career requiring guidance on basic biological design principles; and 3. the broader bioengineering community by providing insight into underlying mechanisms of burden. The paper was mostly well-written, although grammar could be improved throughout, and several of the sections (specifically the Discussion) could be condensed without hampering the message but improving readability. The figures were clear, and the modelling and experiments appeared to be sound. Access to some of the raw underlying data produced would be beneficial (this did not seem to be present in the initial submission) and would help improve the broader impact of the work.

Overall, this paper is a good fit for Nature Communications, but there are some major comments that the authors need to address:

We thank the reviewer(s) for their detailed comments. We have addressed what comments we could as described below. The requests for additional experimental characterization are all highly worthwhile suggestions for future work, but they are beyond the scope of our current study.

Regarding the raw underlying data: Fits for all wells of the microplate assays are provided in supplementary data files/tables. The GitHub repository (and Zenodo archive) contains the raw microplate reader data and analysis/simulation code. DNA sequencing reads were deposited in the NCBI Sequence Read Archive. This information was provided in our initial submission.

1. The analysis of burden didn't cover the combinatorial nature of the constructs assayed. For example, is the burden of a part additive, multiplicative, or follow some other non-linear relationship when put together with other parts?

We would love to address this question, but we are unable to with our dataset.

By analyzing plasmids that had been constructed by many different iGEM teams, we used a "found dataset" akin to that from a "natural experiment" as might be used in an economics study. We would argue that there are some advantages to this approach, in that it provides a relatively unbiased sample of a wide variety of different constructs that are being made for synthetic biology. However, our dataset has limitations relative to this reviewer suggestion and some of the following ones.

It will take future work purposefully assembling combinations of parts to get at how interactions between parts determine the burden of constructs. While there are occasionally some related

BioBrick plasmids with composite parts that share subparts, there are not enough plasmids that systematically vary or combine the same parts to meaningfully examine these interactions in the set of 301 BioBrick plasmids that we analyzed.

We added a mention of this limitation to the Discussion on Lines 678-686 (**bolded text**):

While we were able to establish overall trends that plasmids containing strong constitutive promoters and ribosome-binding sites had a higher chance of exhibiting burden, it was not possible to predict the gene expression component of burden a priori on this set of sequences. **With the limited set of BioBricks we tested, we were also unable to examine whether the burden of individual genetic parts can be used to predict the burden of complex devices constructed from combinations of these parts.** Ongoing improvements in tools for predicting transcription and translation initiation rates and expanding databases of high-throughput measurements^{57,58} may eventually make these types of predictions possible.

This adds to the existing discussion of how interactions may be non-linear on Lines 694-697.

2. Figure 5 looks at single components, those too that representative of broad range of construct types characterized in the work. Furthermore, the analysis of the source of burden was rather limited. It would be useful to discuss potential mechanisms of burden beyond promoter and RBS parts, i.e. could it be related to shifts in metabolism, non-codon optimized genes, non-native components (e.g., not from *E. coli* – as many of the BioBricks in Table 1 seem to be)? Some further comments on this point either here or in the Discussion would be helpful.

Without additional experiments, we believe that it would be premature to speculate further as to why specific constructs we tested exhibit significant burden from gene expression or other sources. It could be related to any or all of the possibilities mentioned by the reviewer. We mention several of these possibilities in the Introduction (Lines 84-95) and Discussion (Lines 687-699). Our Discussion suggests how one might begin to tease out some of these possibilities in the future by building new types of "burden monitor" constructs for examining other cellular capacities that can become limiting for cellular replication, such as protein secretion capacity or protein (mis)folding capacity. Even with such tools or using "omics" experiments, it will likely take significant effort and domain-specific knowledge/experiments to understand why certain BioBricks exhibit "other burden".

3. Plasmid copy number can play a major role in burden, especially if they are hosting complex genetic circuits. Furthermore, vectors with a pUC origin of replication (as used here) can have high variability in copy number. It would be interesting to know how far the assumption of similar copy number for plasmids is held across the constructs tested, particularly those imposing the most burden. Testing a subset of constructs exerting different levels of burden with qPCR or some other method could help to address this question and further verify this assumption.

We agree that these parameters (plasmid copy number and cell-to-cell variation in plasmid copy number) and how they interact with burden would be very interesting to characterize. However, this work would be a significant study in its own right, and our conclusions are agnostic to these details, so we do not plan to characterize them and add any results in this area to our current manuscript.

We added a mention of how burden and plasmid copy number may interact, including with mutation rates, to the Discussion (Lines 753-755):

Second, burden, plasmid copy number, and mutation rates are not necessarily independent parameters and may exhibit cell-to-cell variation.

4. The discussion was too long and lacked deeper insights. I would suggest condensing the content where possible.

We tried to condense parts of discussion (for example, the discussion of the Luciferase construct on Lines 626-637) where this did not remove necessary acknowledgements and citations of work that inspired our study, connections to other ongoing studies of burden for perspective, and mentions of the limitations of our approach and models that could be addressed in future work.

5. These experiments are all carried out in a cloning strain of *E. coli*. How would they generalize to other strains or organisms? Could the model aid in providing some insight into this question? What effects might other *E. coli* strains have (e.g. MG1655, BL21, MDS24)? Further comments on the applicability of the model beyond the context presented would strengthen the utility of the study.

This is another interesting suggestion for a dimension along which our work could be expanded on in the future by us or others. We added a short mention of it in the Discussion (Lines 720-722):

Our experiments were all in a cloning strain of *E. coli*. It would be interesting to examine how burden varies in strains optimized for other applications, such as recombinant protein production.

As many factors may vary in a different *E. coli* host strain, including plasmid copy number; aspects of DNA, RNA, and protein turnover; and mutation rates and spectra (as brought up by another reviewer), we do not feel comfortable speculating about those points. However, our overall conclusion that slowing the growth of a host cell by >45% should make a plasmid unclonable or a genome edit unconstructable should transcend the choice of a specific chassis.

6. It would have been nice to see some experimental application of this knowledge for forward design of new genetic constructs. This is not essential, but it would raise the impact of the work beyond observing existing constructs. Alternatively, some discussion on how the information presented could be used to improve design processes would be useful.

We absolutely agree that a nice follow up study would be to predict the burden of a set of designed combinations of standard parts, create those constructs, and make measurements to test the predictions. It is undoubtedly a limitation in this respect that we used the "found" combinations of parts used by iGEM teams. At best we can advertise this problem, so more synthetic biologists can appreciate the evolutionary implications of burden and understand why they sometimes can't construct plasmids or always acquire mutations in certain plasmids during cloning.

I also had several minor comments that the authors should consider:

Abstract: What are the broader implications of this work? It would be nice to state this at the end to help the reader understand the effect this work will have.

We added an overall summary statement to the end of the Abstract:

Our results establish a fundamental limit on what DNA constructs and genetic modifications can be successfully engineered into cells.

Abstract: It may be helpful to include the URL for the online simulation tool.

Thank you for the suggestion. We added a stable, short URL pointing to the online version of the simulation tool to the Abstract.

L41: “in new and more challenging environment” -> “into increasingly challenging environments”

We prefer our phrasing. We are purposefully highlighting "new" environments alongside "more challenging" environments, versus always the combination of the two.

L45: “potentially making their functions unpredictable and unreliable” – perhaps be explicit that evolution can cause cells to modify functions and lead to a breakdown in function?

The next few sentences explain and elaborate on this statement (Lines 47-55).

L54: “constructs must use” -> “constructs use”

We deleted "must" as suggested.

L56: Reference 14 is not used correctly, Rouche et al. demonstrated that plasmid copy number does play a role in overall cell burden, especially high copy plasmids.

We agree this citation is not appropriate here. Thank you for catching that!

We wanted to communicate that the fitness cost of copying DNA added to a cell is typically negligible (small in magnitude) compared to the cost of the genes expressed from that DNA.

The Rouche *et al.* paper reports a 0.065% growth rate reduction per plasmid copy from analyzing a library of pUC19 copy number mutants. They interpret this cost as arising from DNA replication because "pUC19 is 2686 bp in size, which is approximately equal to 0.058% the length of the *E. coli* genome". From a wider reading of the literature, we believe this simple model is probably not the right way to interpret their results. It neglects that each pUC plasmid copy is expressing β -lactamase and encodes the lacZ α fragment under control of P_{lac} . Despite the host cells expressing LacI from the chromosome, there is known to always be some level of leaky expression.

Other papers suggest that neither DNA polymerases nor dNTP pools should be limiting, even when there are hundreds of copies of plasmids in cells. However, we don't know that this any more than conventional wisdom. We couldn't find any studies that do quantify burden from just DNA replication in cells and find it to be negligible. One challenge is that it is difficult to make a DNA sequence of any significant size that is completely inert as far as transcription yet maintains fairly normal properties in terms of nucleotide composition. Perhaps another idea for a future study!

In the end, we decided to remove the original sentence entirely and also the citation (though it is a fascinating study that uses very nice methods to examine and manipulate plasmid copy number).

L59: “synthesize RNAs and proteins” -> “synthesize RNAs and proteins that are not native to the cell”

We incorporated this suggestion.

L59: “Protein overexpression...” – very long sentence that needs to be split.

We split this sentence and reworded the remainder of the second half to make it shorter.

L64: There are some earlier references to monitoring burden using combined sequencing approaches that are missing: 10.15252/msb.20188719, 10.15252/msb.20167461

We added the first of these references here, since it is the most recent and includes Ribo-seq data, as we are describing here. The other is cited later. Thank you for pointing these out.

L85: “It has been proposed that genetic reliability...” – what do you mean? Could you elaborate a little on what a “certain level” refers to and the issues in defining this property. The allowable fluctuations in function are typically very application dependent so reliability has varying importance.

By "certain level" we meant that a minimum amount of a measurable output like GFP expression, biosensor dynamic range, or product titer remains when measured in bulk for the entire cell population. As the reviewer suggests, the requirements for genetic reliability in practice can be very application dependent. It would take quite a bit more explanation to describe specific types of applications and when/where genetic reliability impacts them. That seems more appropriate for a review on the topic, versus the point we are making in this paragraph, so we did not expand our discussion. To simplify, we eliminated mentioning "certain level" in this phrase (next comment).

L86: “sense of for how” -> “sense of how”

We changed the phrasing here to "in the evolutionary sense of how many cell doublings it takes for an engineered function to decay within a population".

L102: “other, currently unknown reasons” -> “other, currently unknown, reasons”

We think our comma usage is appropriate, but perhaps there are different styles.

L104: “evolutionary failure can be used to improve the reliability of bioengineering” – is that really what is shown in this work? I would suggest that these large studies are hugely important for understanding the stability of parts and this information *can* be used to aid in the design of more robust synthetic biology.

We changed this sentence to be more forward-looking as far as the "improve" part (Lines 125-127):

Our work demonstrates standardized frameworks for measuring burden and simulating the dynamics of evolutionary failure that can be used to improve the reliability of bioengineering.

L110: “alleviate a burden” -> “alleviate the burden”

We appreciate the suggestion but prefer our original phrasing since there may be multiple types of burden from one construct.

L112: “will outreplicate and” -> “will replicate more quickly and” – you might also want to use the term “competitive exclusion” which is technically what is happening.

Thank you for the suggestion, but we think "outreplicate" is used widely enough in the scientific literature to justify keeping this as is.

L132: “would be likely to lead to” -> “would likely lead to”

We appreciate the suggestion but prefer our original phrasing.

L144: “plays into the mutation rate” -> “affects the mutation rate”

We changed it to “affects mutation rates”.

L147: I'm not sure what you mean by "more densely coded". I assume you are alluding to the degeneracy in the genetic code or robustness to mutations. It would be helpful to more explicitly state that.

Yes, we are referring to robustness to mutations as a continuation of the description of information content in the previous sentence. We changed the phrasing to "are less robust to base changes".

L153: Please include some other recent examples of mutational hotspots: 10.1038/s41467-021-26286-9

We added the suggested citation and made our statement about hotspots more general to include mechanisms that are not due to sequence repeats, since a hairpin structure seems to be causing the hotspot in this case. Our original citation here includes a sort of minireview of quantitative studies of mutational hotspots in microbes because it uses them to calibrate models used by a software tool. We couldn't find more recent studies that add much to that literature. There have been nice studies of smaller biases in mutation rates (e.g., due to DNA curvature or sequence context), but the magnitude of these biases is usually small (<10-fold and usually less than <2-fold) compared to what we would classify as a hotspot and want to bring attention to with this statement.

L163 L164, L170: Can you please include the units of the rates you are referring to. Please check throughout and always include the units with any measurement values in the main text.

All the units in this paragraph are mutations per cell division. We added an additional spot where we repeat the units but think that it becomes unnecessarily wordy to write "mutations per cell division" after every mention of a value in this paragraph. The "mutation" part is also implicit in places.

L177: "these "jackpots" occur" -> "these events occur"

We changed this as suggested.

L186: "early jackpots" – I'd suggest not using the term "jackpot". Also, in this context, how can there be a mutation earlier than when the simulation starts. Is it not the fact that multiple mutations are clustered early on, which is unlikely to happen and so rarely seen?

Mutational jackpot events occur when there is a mutation early in the growth of a culture, such that it leaves a much higher number of mutant offspring than when a mutation occurs near the end of growth and only gives rise to one or a few mutant offspring by the time the culture stops growing. The concept and term go back to the classic Luria-Delbrück experiment (fluctuation test).

We added this additional explanation to our statement (Lines 237-238) . It now reads: "due to jackpot events when a mutation occurs early in the growth of a population". It is not quite the situation described by the reviewer of multiple mutations early in the same simulation. Only one is required.

We are unsure what the reviewer means by "how can there be a mutation earlier than when the simulation starts." This paragraph refers to mutations that appear immediately at the beginning of the simulation, but never before it begins. In a deterministic ODE model with a mutation rate of 10^{-6} , you immediately have " 10^{-6} mutant cells" in your population after the first simulated cell division, and this "subpopulation" will begin expanding because it is more fit. It does not respect whole numbers.

L197: "represent the output" – is that a mean of the simulations? If so, where is the variability?

This plot shows the cumulative distribution function (CDF) for the time that it took each of 10,000 different simulations to reach 50% failed (mutated) cells in the population. It shows the entire distribution of this key per-simulation summary statistic. If a curve passes through a fraction of 0.1 at 50 cell divisions, this means 50% or more of the population was mutated cells in 1,000 of the 10,000 simulations run with that set of parameters by 50 cell divisions or sooner. The median simulation time to 50% failure is where the curve passes a fraction of 0.5. The variability between simulations is displayed by the shape of the curve.

We edited the figure legend to try to more clearly explain what is being shown and how to interpret it:

Fig. 3. Cumulative distributions of times to 50% failure in stochastic simulations. Each curve shows the values of this summary statistic of failure for 10,000 simulations with a given parameter combination. More variability in the time to 50% failure leads to a flatter curve.

L208: "We created..." – Some of the discussion in this paragraph would be better suited to the Discussion, not the Results.

We moved and merged parts of this with the final paragraph of the Discussion (Lines 761-788).

L219: "evolutionary constraints predicted" -> "evolutionary dynamics predicted"

We mean "constraints" here. Our experimental data does not speak to the "dynamics". This would mean tracking what percent of different populations had mutated plasmids over time.

L265: "significantly decreased E. coli growth" – I don't understand what you mean by significantly here: statistically significant to the control (may not be very impressive in absolute terms) or seeing a drop greater than some threshold (perhaps more impressive). Could you please be explicit and avoid terms like significant unless explained. In many cases the term "significant" doesn't really add anything and could be omitted and actual changes stated. Please consider this comment throughout the manuscript.

We understand the reviewer's reading of "significantly decreased" as "exhibited a large decrease", but we consistently use "significant" here and elsewhere only in the statistical sense of being able to reject a null hypothesis. We believe everything is reported according to established conventions.

We double-checked our usage elsewhere in the text, and we did follow the reviewer's advice to reduce how many times we describe plasmids as having "significant burden" when appropriate. We now state a statistical test and *p*-value threshold for rejecting the null hypothesis every time that we use the word "significant", usually at the end of the same sentence in parentheses.

In this specific case, the next few sentences address what the reviewer suggests is needed for context, which is to examine the magnitude of the effects. Here, we use "significantly greater than" because simply saying that *N* BioBricks had a burden >Y% (or a similar statement) would not account for uncertainty due to measurement errors in the values we are comparing.

L329: Studies of terminators have shown they are not always perfect and are sometimes highly context dependent. It would be worth mentioning that with some evidence. 10.1038/nmeth.2515, 10.1038/s41467-022-28074-5

As suggested, we elaborated on this point and cited these two studies.

L343: Would you not expect to see some mixed colonies? How was the plasmid DNA prepared for

sequencing, was clean/reliable data quality seen across entire plasmid sequence? (I believe this information is provided by the full-plasmid sequencing companies)

This project was performed in the days before on-demand nanopore plasmid sequencing services were widely available. We performed our own Illumina sequencing of plasmid minipreps (as described in the Methods and our SRA submission). We analyzed Illumina data using our *breseq* pipeline that does report "polymorphic" mutations in mixed samples, a capability that we and others have demonstrated in many prior studies of bacterial genome evolution. We report results only for plasmids that had sufficient coverage to reliably call mutations across their entire sequences.

We thought we might see mixed populations of cells with mutated and unmutated plasmids, as the reviewer suggests, particularly for more burdensome plasmids. However, we did not see any convincing cases of catching these dynamics as escape mutants were taking over a population. Without performing time-course sequencing of evolution experiments, one does expect that you have to be "lucky" to catch these mutational sweeps when they can most reliably be called as polymorphisms (in the 20-80% frequency range) from next-generation DNA sequencing data.

L355: Does this have implications for how DNA is verified before submission to the registry? This might be nice to add to the Discussion.

Thank you for this suggestion. Ideally, all iGEM teams (and synthetic biologists in general) would sequence the plasmid stocks they send to and receive from the iGEM Registry (or other repositories, such as Addgene). Cheaper whole-plasmid sequencing services are making this more accessible.

We added this point to the Discussion along with citations to two opinion pieces that call for fully determining and reporting the sequences of plasmids used in synthetic biology (10.1038/nbt.1753 and 10.1371/journal.pbio.3002376) on Lines 609-613:

Our findings support calls for researchers to report the full sequences of plasmids they create and submit to repositories such as the iGEM Registry^{55,56} and caution that one should also verify the sequences of plasmid stocks obtained from repositories. This information will make it possible to recognize when evolution is undermining DNA constructs and experimental results.

L390: "cell for expressing" -> "cell to express"

We appreciate the suggestion but prefer our original phrasing.

L446: "reduced in direct proportion" -> "reduced proportionally"

This text was removed in response to a comment by another reviewer.

L483: "performing simulations" -> "performing computational simulations"

We changed this as suggested.

L497: "their burdens" -> "their burden"

We changed this as suggested.

L555: "rate of E. coli host cells" -> "rate of the host E. coli cells"

We appreciate the suggestion but prefer our original phrasing.

L602: "Researchers designing..." – very long sentence that needs to be broken into several.

We broke this sentence up in the rewrite of the last paragraph of the Discussion.

L610: "do not burden their growth by more than ~30%" – this sentence doesn't make sense, what is burdening growth by a percentage. Perhaps reword to be more explicit, e.g., "ensure your constructs do not lead to more than a 30% drop in normal growth rate"

We agree this was poorly phrased. We reworded it to: "... rule of thumb: to avoid the specter of unwanted evolution, don't attempt to engineer a microbial cell in a way that slows its growth rate by >30%." This phrasing includes cells engineered with gene edits instead of just DNA constructs.

L683: the authors mention dephosphorylating their DNA digest products using calf intestinal phosphatase, which was followed by gel extraction and ligase with T4-ligase. However, dephosphorylated DNA cannot be ligated without a phosphorylation step, such as by using T4-polynucleotide kinase. Is this protocol correct?

Our methods describe that the mTagBFP "insert" in our cloning scheme from plasmid K592100 was only digested (not dephosphorylated), so it retains a 5' phosphate. The pSB1C3 promoter+RBS composite part BioBrick plasmids that we use as "vector backbones" were digested with other restriction enzymes so they serve as the recipient plasmids during BioBrick assembly. They were dephosphorylated to prevent self-ligation. T4 ligase can join the phosphorylated mTagBFP insert to a dephosphorylated pSB1C3 promoter+RBS vector backbone. We added "insert" to our description of the mTagBFP sequence to make it clearer that it was not also a "vector backbone", which seems to have been the source of this confusion.

L713: Experimental cultures were prepared directly from "-80 °C" stocks – could this affect the results, i.e., starting with a mixed colony?

This is an important issue. If the question is whether this is the *best* way to conduct these types of experiments versus picking a single colony from the freezer stock for each biological replicate, one could make arguments on both sides. It is true that the inoculum from the frozen stock could already have genetic diversity that arose during growth of that culture from a transformant colony (some cells could have mutated plasmids). If, on the other hand, one picks a single colony by streaking out the freezer stock (or freshly transforms plasmid into cells) right before doing a microplate assay, then one increases the risk of starting a population that derives wholly from a cell with a mutated plasmid. If the plasmid encodes GFP, then one could theoretically screen for which colonies are unmutated; however, most of the BioBricks do not have screenable markers. In the ideal experiment, one might do many technical replicates from each transformed clone or sequence the plasmids in cells from every biological replicate to know what "noise" was due to measurement error versus mutations.

We had to factor in what was practical for the humans doing the experiments, in terms of time and materials, so we went with the methods we describe for this study. If mutants were present at appreciable frequencies in the revived inocula, we would expect this to only occur for plasmids with very high burden, and it would lead to our measurements underestimating burden, as cells in the population would have a mixture of intact plasmids and plasmids that mutated to reduce burden.

Figure 1, 2 and 3: It felt like several of these could be combined.

We appreciate the suggestion but prefer our original figure layout.

Figure 4: Please provide some representative growth curves for the data presented in Figure 4; this

is not necessary for all the constructs, just a small set of samples spanning the range of burden observed to be sure there are no strange growth effects.

We added this as a new supplementary figure (**Supplementary Fig. 1**).

Prior to our initial submission, we did visually check our curve fits (all of them) to ensure that there was nothing strange going on within the windows used to determine growth and fluorescent protein production rates. This check resulted in eliminating runs of some plates from the results we report. (The raw data for even these "failed" runs is available in our GitHub repository / Zenodo archive).

Figure 4B and 7C: Both data sets presented in these figures display growth rate for the iGEM constructs and for the BFP plasmids, as well as discuss the "significance" level of the burden imposed on the cells. The colour scheme used to highlight the "significant" burden plasmids is different between the two figures, being yellow-orange and pink for 4B and 7C, respectively. For consistency, it would be helpful for the colours to be the same so that it is easier to draw the link between the two datasets.

This is a great suggestion. We changed the color scheme in **Fig. 7C** to match that in **Fig. 4B**.

Figure 6A and 6B: These could be combined into one panel.

We appreciate the suggestion but prefer our original figure layout.

Figure 6C: The relative intensities for the BFP and RFP expression are very difficult to discern from the colour gradient alone. It would be helpful to provide the BFP and RFP fluorescence intensity data on a separate graph, either in the main manuscript or Supplementary Information.

We added **Supplementary Fig. 6** to show the BFP/RFP values for each plasmid.

Reviewer #1 (Remarks on code availability):

The data and code was well organised and commented. I have not been able to run all the code present, but it contains sufficient instructions and looks to be of a good standard. Furthermore, the online application is functional and a joy to use.

Thank you for this comment. As we were not quite fully in line with journal policies, we expanded the documentation a bit more. One major update was adding instructions for running the interactive burden simulation tool locally on a user's machine so they do not have to rely on our web version.

Reviewer #2 (Remarks to the Author):

We appreciate the time you put into your co-review.

Reviewer #3 (Remarks to the Author):

Opinion on Radde et al., "Measuring the burden of hundreds of BioBricks defines an evolutionary limit on constructability in synthetic biology"

The authors investigated the the effect of mutation rate and growth retardation, i.e. burden on the

stability of genetic parts and devices propagated on plasmids in *E. coli*. They generated two tunable models, a deterministic and a stochastic to predict the loss of the correct genetic circuit upon culturing. As a rule of thumb, they declare that a burden > 45% leads to unclonability (loss of function already at the stage of colonies obtained after transformation), and recommend to keep the burden below 20-30% if upscaling to 1000 L fermentors is planned.

Next, they measured the burden of 301 BioBricks parts elicited on *E. coli*, and found 59 to display a >10% burden with high confidence. As expected from the model, no construct displayed a burden >45%. In addition, they found that two of their control plasmids strongly expressing BFP had gone through a mutant selection that reduced their burden from 45.8% to 17.8% and 41.9% to 17.2%, respectively, further supporting that burdens approaching or exceeding 45% will lead to the loss of the correct construct from the bacterial population. The authors found that strong constitutive promoters and strong ribosome binding sites significantly increased the chances of high burden, but specific predictions for the burden of individual constructs could not be made based on this set of data.

Following, the authors used a reporter fluorescent protein, GFP expressed from the chromosome to measure the effect of the cloned genetic construct on the transcription/translation capacity of the host cell. They found that in nearly all cases, the reduction in growth rate corresponded to the reduction of GFP expression, therefore the burden could be attributed to the diverting of the transcription/translation machinery to express the cloned construct. The authors pinpointed a few exceptions, where they provide numerical data for the fraction of the burden caused by other, unknown mechanisms (bo/b).

The Discussion provides valuable recommendations for the scientific community to reduce the instability of clones, and/or the presence of erroneous plasmids at plasmid repositories. The authors also give good ideas for future research, e.g. the development of further burden monitors, or carrying out high-throughput gene expression measurements to train AI-based predictors of transcription and translation initiation rates.

Overall, the text is reader-friendly, clearly written in perfect English. The selected methods are sound; the analysis of the data applies sufficient statistical analysis. The experiments are well described, and the conclusions are well supported.

Thank you for the positive comments and taking the time to provide feedback on our manuscript.

Major remark:

The manuscript practically disregards the fact that the burden and the mutation rate are not independent variables. It has been displayed that various stressors, including that of transgene expression elevates the rate of point mutations, deletions and insertions via inducing error-prone polymerase genes, RecA and the transposition of mobile genetic elements, respectively. Even a gene as inert as GFP has been shown to elevate mutation rates 1.5-2 fold, when expressed. I understand that the model was to be kept simple, but this shortcoming could be mentioned in the discussion. The use of engineered bacterial strains displaying reduced mutation rates for cloning purposes is cited. Without overrating these strains, it is worth adding that this phenomenon increases their advantage compared to that inferred from mutation rates measured under unstressed conditions (e.g. Microbial Cell Factories 2012, 11:11).

This is an important point. Thank you for mentioning it. We added a citation to this study after a new sentence describing how burden and mutation rate are not necessarily independent (Lines 753-755):

Second, burden, plasmid copy number, and mutation rates are not necessarily independent parameters and may exhibit cell-to-cell variation. For example, overexpression of recombinant proteins can activate stress responses and increase mutation rates.⁷³

We also added a citation to the same study earlier in the Discussion when mentioning the benefit of engineered reduced-mutation strains.

Minor remarks:

Line 180: “deterministic simulations consistently overestimate how unstable a construct will be for a given combination of parameters”. Is there evidence that the deterministic models truly overestimate compared to experimental data? If not, I think the authors should add “compared to stochastic models”.

The reviewer is correct. We made the suggested change to make the sentence accurate.

Line 387, line 431: the authors refer to a “proportional” reduction of gene expression capacity and the growth rate. I would avoid the word “proportional”, unless the fitted line on the correlation plot passes through the origin. On Figure 6, neither of the two lines (especially not the RFP trendline) passes through the origin. The linear correlation is nevertheless evident.

We agree that it would be incorrect to say that the growth rate is proportional to the remaining gene expression capacity of a cell, which would imply linear fits passing through the origin. However, our statements are all about the *reduction in gene expression capacity* (GFP production rate) being proportional to the *reduction in growth rate*, which only implies a linear relationship passing through the “point of no burden” at normalized coordinates of (1, 1). We kept the current description, as we feel it is the most straightforward way of verbally explaining these results and connecting to the logic of how burden from other sources will cause deviations from this proportional reduction.

Reviewer #4 (Remarks to the Author):

The authors of this manuscript aim to quantify, both theoretically and empirically, the well-known evolutionary instability challenge in synthetic genetic constructs. The authors first theoretically examine the consequences of the interaction between mutation rate and decreased growth rate, drawing strong and useful quantitative predictions from this simple model. These predictions are then confirmed, leveraging the iGEM parts collection to obtain a large and diverse but highly comparable collection of synthetic constructs, producing specific data, guidance, and protocols that are likely to be of use to many researchers in the future.

Overall, this manuscript was admirably clearly presented and a pleasure to read. I see only a few minor points that need cleanup before its publication:

- p8: It would help to make it clear that the iGEM plasmids are pUC derivatives at this point

We don't think it makes sense to add this detail in the middle of the modeling section, which is well before we introduce that we will be studying plasmids from the iGEM Registry in the Results. We already have this detail in the first paragraph of the “Burden of BioBrick Parts” Results sub-section.

- p8: Are you able to numerically estimate mutation rate in the constructs that you measured? It would strengthen things if you could, but I didn't see that in your data.

Unfortunately, this is not possible, which is why we discuss mutation rates more generally in terms of ranges of values and examine their effects in the simulations within a few plausible orders of

magnitude. In general, the per-base point mutation rates in these plasmids are probably similar to those we have measured in a pUC plasmid (in the cited study 10.1093/nar/gky751), but there is likely to be significant variation in the effective rates for "failure" mutations in plasmids, due both to differences in their sizes and information content and because some may have mutational hotspots.

- Fig 2: use line patterns (e.g., dashes) to make blue lines more distinguishable

We didn't make this change because these "signposts" are always at the same values, and we want them to look the same in Figs. 2 and 3. We made the red lines in **Fig. 3** dashed (next suggestion), which we thought was more important, and having both sets dashed in **Fig. 3** looked too busy.

- Fig 3: use line patterns or markers to make red lines more distinguishable

We changed the figure to use line patterns for these curves. We think that it also helps distinguish that different types of data are being graphed with red lines in **Fig. 3** versus **Fig. 2**: curves that summarize the results of many simulations versus curves for individual simulations, respectively.

- p12: How were the 301 parts chosen? Why do exactly 3 "other plasmid" biobricks? Why the relative balance of pSB1C3 vs. pSB1A2? Was this just all the parts you had available, or was there some other logic behind including some and omitting others?

The pSB1C3 versus pSB1A2 balance was not planned. The 301 parts we analyzed are those that "survived" our pipeline. It began with undergraduates on our iGEM team transforming as many plasmids as they could from the distribution kit. Then, they made plate reader measurements of burden, not all of which yielded usable data. The 3 BioBricks on other plasmid backbones arose when we analyzed sequencing data for our plasmid stocks and realized that the information in the distribution kit about those plasmids was not accurate: they were annotated as being on pSB1C3 or pSB1A2 backbones. Fortunately, these cases of incorrect annotation seem to be rare. It's likely they are mistakes in the information provided by the iGEM teams that submitted these plasmids, as the antibiotic resistance marker is the same in all cases as the one on the annotated backbone. These BioBricks on the "other plasmid" backbones were not outliers in terms of their burden.

- p13: Benjamini–Hochberg is likely to be unfamiliar to many readers (including myself): please add a citation, even if it's to a textbook

We added a citation to the original paper (10.1111/j.2517-6161.1995.tb02031.x) on Line 346 where we first mention this correction. It is a standard statistical method to use with high-throughput data.

- p16: The case of K880005 doesn't seem surprising - even if there's no ORF, the promoter and RBS will still be heavily recruiting transcription factors and ribosomes. Moreover, if there are small peptides produced before the promoter, these may be toxic for the cells, as seems to be the case for BCD promoters.

We think some readers may find this result surprising, so we feel it is worth noting. We agree with the reviewer's interpretation of what is likely causing burden in this case. Our description mentions that the burden may result from transcription/translation across the BioBrick suffix into the backbone.

- Fig 6A: it would be good to use the proper SBOL visual symbol for chromosomal integration. See example with the glyph definition at: <https://github.com/SynBioDex/SBOL-visual/tree/master/Glyphs/SequenceFeatures/chromosomal-locus>

Thank you for pointing out that our SBOL visual symbol was not quite right. We changed it.

- Fig 7C: How do you explain the increased gene expression capacity of K523014 and the gene expression impact higher than growth impact for the few "far below the line" pink triangles? If these are predictable based on construct contents, it would be good to discuss. If these are unexplained variation, then it is not clear whether the other "pink square" points should be considered significant, since there aren't many more of these than there are these anomalies in non-predicted directions.

The fraction of "other burden" in the case of K523014 has a 95% confidence interval of 1.04–1.98 (shown in **Table 1**). If it is >1.0, then it would indicate that the BioBrick "increased" gene expression capacity. It is trending that way, but not significantly so after a correction for multiple testing.

The reviewer's observation that there are quite a few points in non-predicted directions, such as less burdensome than predicted from the reduction in gene expression capacity, is valid. Overall, our measurements of both growth rates and fluorescent protein production rates were noisier than we would have liked, which is a reason that we do not comment in more detail about the results for specific BioBricks and speculate about why certain ones exhibit significant "other burden".

That said, K523014 and the other BioBricks shown as pink squares in **Fig. 7C** (now orange squares in the revised version) do pass a rigorous statistical test for being above the diagonal relationship that is expected if all of their burden was due to depleting gene expression capacity (see **Methods**). There are also more points that are above this line than below it among the >10% burdensome BioBricks (pink, now orange, triangles), which supports our overall interpretation.

The estimated 95% confidence intervals on the "other burden" fraction are provided for all of our constructs in **Table S3**. This information and the rates fit for each individual experimental replicate in **Table S2** can be used to see the level of noise in our measurements for any specific BioBrick.

- p26: typo - "measurement high-throughput"

We corrected this to "measurements high-throughput".

Reviewers' Comments:

Reviewer #1:

Remarks to the Author:

We thank the authors for their careful consideration of our comments. Overall, we are happy to support for publication in this form but were a little disappointed that they had not made some effort to address our major comments with the data they had available, broadening the findings beyond how burden is typically assessed. While we agree that additional experiments would be required in several cases to make this rigorous, even some basic analysis would have been interesting. E.g. for comment (2) regarding the sources of burden, it would have been feasible to assess the source organism of parts to see if there was any connection with where a part had originated.

Reviewer #2:

Remarks to the Author:

Reviewer #3:

Remarks to the Author:

Thank you for the corrections and the explanation. I have no further points of criticism.

Reviewer #4:

Remarks to the Author:

I am satisfied with the authors responses to the points I raised.

Reviewer #1 (Remarks to the Author):

We thank the authors for their careful consideration of our comments. Overall, we are happy to support for publication in this form but were a little disappointed that they had not made some effort to address our major comments with the data they had available, broadening the findings beyond how burden is typically assessed. While we agree that additional experiments would be required in several cases to make this rigorous, even some basic analysis would have been interesting. E.g. for comment (2) regarding the sources of burden, it would have been feasible to assess the source organism of parts to see if there was any connection with where a part had originated.

As suggested by the Reviewer and requested by the Editors, we examined whether there was any correlation between the source organism of a BioBrick and its burden.

Methods Summary: we manually curated the source organisms of the 301 BioBrick parts for which we measured burden by examining the descriptions on the iGEM Registry web pages and conducting further BLAST and literature searches. Then, we tried three ways of categorizing the source organisms with the goal of that grouping them into broad categories that reflected their taxonomic similarities and differences from *E. coli*. We explored further variations on this classification scheme that optionally omitted either or both of two types of features (non-protein-coding features and fluorescent proteins) when assigning the organism of origin.

For all three categorization schemes and all four combinations of including or not including each of the two omissions (12 total datasets), we performed two tests for trends in burden with respect to BioBrick source organism: (1) We used (non-parametric) Kruskal-Wallis tests to see whether the normalized growth rates of *E. coli* transformed with the BioBrick plasmid differed with the source organism category; (2) We used likelihood-ratio tests comparing binomial regression models to test whether the chance that a BioBrick was classified into the set of 59 burdensome BioBricks differed with the source organism category.

Results Summary: Of the total of 24 analyses (two statistical tests crossed with 12 datasets with different classifications of organism of origin of BioBrick parts), just one yielded a p -value < 0.05 for rejecting the null hypothesis that there was no significant effect of source organism on BioBrick burden. That one combination, the four-category scheme with no omissions tested with the binomial model, was only barely below this significance threshold ($p = 0.047$).

Overall, we did not find any compelling trends in the burden of BioBricks with respect to the taxonomic classifications of the organisms in which the sequences of their genetic parts originated. Given that the BioBricks in our dataset and their constituent genetic parts have very diverse functions that vary across the different source organisms, this is probably not very surprising. Perhaps one would expect a trend like the one the Reviewer #1—in which parts from certain organisms were more likely to have burden—if the *same gene* was systematically transplanted from many different source organisms into *E. coli*. It is also possible that a signal would be detected in a larger dataset or if one could deconvolute the effects of both cryptic and known gene expression signals from any organism-of-origin effect.

Updates to Manuscript: We added the negative result of this analysis of organism of origin to the revised manuscript in the new **Results** section "**BioBrick burden is not correlated with organism of origin**", new **Methods** section "**BioBrick source organism analysis**", new **Supplemental Fig. 4**, and new **Supplemental Table 1**. We also updated **Supplemental Data 3** to include organism of origin columns. Finally, we added a sentence to the **Discussion** to put this finding in context by citing a high-throughput study of horizontal gene transfer that examined why certain genes from 79 prokaryotic genomes were "unclonable" in *E. coli*. This study also found that gene expression was more important than relatedness to *E. coli* for explaining cloning success, in agreement with the findings of our new analysis.

These changes are highlighted in yellow in the "Revised Manuscript with Highlighted Changes" related manuscript document.

We also added source organism classification metadata and the analysis script used to perform these statistical tests and generate these plots to our code that is on GitHub and archived with Zenodo with the DOI provided in our submission.

Reviewer #1 (Remarks on code availability):

The code and data are of a sufficient standard.

Reviewer #2 (Remarks to the Author):

Reviewer #3 (Remarks to the Author):

Thank you for the corrections and the explanation. I have no further points of criticism.

Reviewer #4 (Remarks to the Author):

I am satisfied with the authors responses to the points I raised.